



# Spatiotemporal patterns of the fossil-fuel $CO_2$ signal in central Europe: Results from a high-resolution atmospheric transport model

Yu Liu[1,2], Nicolas Gruber[1,2], and Dominik Brunner[3]

[1]Environmental Physics, Institute of Biogeochemistry and Pollutant Dynamics, ETH Zurich, Zurich, Switzerland
[2]Center for Climate Systems Modeling (C2SM), ETH Zurich, Switzerland
[3]Laboratory for Air Pollution/Environmental Technology, Swiss Federal Laboratories for Materials Science and Technology, Empa, Duebendorf, Switzerland

*Correspondence to:* Nicolas Gruber (nicolas.gruber@env.ethz.ch)

**Abstract.**

The emission of $CO_2$ from the burning of fossil fuel is a prime determinant of variations in atmospheric $CO_2$. Here, we simulate this fossil fuel signal together with the natural and background components with a regional high-resolution atmospheric transport model for central and southern

Europe considering separately the emissions from different sectors and countries on the basis of emission inventories and hourly emission time functions. The simulated variations in atmospheric $CO_2$ agree very well with observation- based estimates, although the observed variance is slightly underestimated, particularly for the fossil fuel component. Despite relatively rapid atmospheric mixing, the simulated fossil fuel signal reveals distinct annual mean structures deep into the troposphere

reflecting the spatially dense aggregation of most emissions. The fossil fuel signal accounts for more than half of the total (fossil fuel + biospheric + background) temporal variations in atmospheric $CO_2$ in most areas of northern and western central Europe, with the largest variations occurring on diurnal timescales owing to the combination of diurnal variations in emissions and atmospheric mixing/transport out of the surface layer. Their co-variance leads to a fossil-fuel diurnal rectifier effect

with a magnitude as large as 9 ppm compared to a case with time-constant emissions. The spatial pattern of $CO_2$ from the different sectors largely reflects the distribution and relative magnitude of the corresponding emissions, with power plant emissions leaving the most distinguished mark. An exception is southern and western Europe, where the emissions from the transportation sector dominate the fossil fuel signal. Most of the fossil fuel $CO_2$ remains within the country responsible for the

emission, although in smaller countries, up to 80% of the fossil fuel signal can come from abroad. A fossil fuel emission reduction of 30% is clearly detectable for a surface-based observing system for



atmospheric $CO_2$, while it is beyond the edge of detectability for the current generation of satellites with the exception of a few hotspot sites. Changes in variability in atmospheric $CO_2$ might open an additional door for the monitoring and verification of changes in fossil fuel emissions, primarily for
surface based systems.

## 1   Introduction

With annual $CO_2$ emissions from fossil fuel burning and cement production having soared in the recent decades and approaching 10 Pg C $yr^{-1}$(Raupach et al., 2007; Friedlingstein et al., 2014; Le Quéré et al., 2016), these fluxes have reached the same order of magnitude as the natural ex-
change fluxes between the atmosphere and land surface and between the atmosphere and the ocean, respectively (Sarmiento and Gruber, 2002; Le Quéré et al., 2016). Thus, the fossil fuel emissions have become a key driver for the spatiotemporal dynamics of atmospheric $CO_2$, not only close to major sites of emissions, but also far downstream (Peylin et al., 2011; Keppel-Aleks et al., 2013; Nassar et al., 2013). This represents simultaneously a challenge and an opportunity. It is an opportunity
since the substantial and growing size of this fossil fuel $CO_2$ signal facilitates the use of variations in atmospheric $CO_2$ to monitor and verify changes in fossil fuel emissions (Bovensmann et al., 2010; Velazco et al., 2011; McKain et al., 2012; Ciais et al., 2014; Shiga et al., 2014). At the same time, the large fossil fuel $CO_2$ signal complicates the use of atmospheric $CO_2$ observations to determine sources and sinks of $CO_2$ driven by the land biosphere through atmospheric inverse modeling meth-
ods. This requires the separation of the biospheric signal in atmospheric $CO_2$ from the total signal, which is usually accomplished by subtracting an estimate of the fossil fuel component from the measured atmospheric $CO_2$ concentration. This implies that any error in the fossil fuel component tends to be projected directly onto the inversely estimated biospheric fluxes (Nassar et al., 2013; Peylin et al., 2011). Thus, in order to benefit from the monitoring and verification opportunity as well as to
minimize the magnitude of the challenge associated with atmospheric inversions, it is paramount to well characterize the fossil fuel component in atmospheric $CO_2$ in time and space.

     Two sets of approaches have been developed to determine this fossil fuel component in atmospheric $CO_2$. A first set of approaches relies on concurrent observations of carbon monoxide (CO) and/or radiocarbon to determine the fossil fuel component in the observed atmospheric $CO_2$ varia-
tions (Bréon et al., 2015; Ciais et al., 2013; Levin and Karstens, 2007; Van Der Laan et al., 2010; Turnbull et al., 2011; Newman et al., 2013; Vogel et al., 2013; Lindenmaier et al., 2014; Vardag et al., 2015; Oney et al., 2016, in review). A major advantage of these observation- based methods is that they do not require any atmospheric transport modeling, and thus are not sensitive to any errors in the modeled atmospheric transport. A major disadvantage is that these observation-based estimates
are available only at a relatively small set of observing sites, providing a very limited picture of the spatiotemporal dynamics of the fossil fuel signal for larger areas. Further complications may arise





from e.g., poorly known and varying ratios of the emissions of CO and $CO_2$ in the case of CO-based methods (Oney et al., 2016, in review), or the emission of radiocarbon from nuclear power and reprocessing plants in the case of radiocarbon-based methods (Graven and Gruber, 2011).

In the second set of approaches the fossil fuel $CO_2$ signal is modeled, starting from the specification of fossil fuel emissions as a bottom boundary condition in an atmospheric transport model, and then running this model forward in time (Peylin et al., 2011; Ogle et al., 2015). A key advantage of this set of approaches is that the spatiotemporal dynamics is fully resolved. But this comes at the disadvantage that the resulting accuracy of the modeled fossil fuel $CO_2$ signal not only depends on

the quality of the fossil fuel emissions data, but also on that of the transport model. The latter disadvantage is well illustrated by the results of a recent model intercomparison study, where inter-model differences in the simulated spatiotemporal pattern of the fossil fuel $CO_2$ were 2-3 times larger than the differences resulting from the use of different emission inventories (Peylin et al., 2011). Of particular relevance is the resolution of the atmospheric model, as this is key to better resolve the

topography and land surface contrasts that govern much of the atmospheric circulation and mixing in the lower atmosphere.

The challenge associated with the modeling of atmospheric transport is particularly acute for the fossil fuel component, since fossil fuel emissions are distributed in time and space in a highly heterogeneous and non-Gaussian manner (Ray et al., 2014). This reflects the nature of the processes

underlying these emissions, ranging from the point source nature of the emissions from coal-fired power plants, whose emissions vary in response to changing needs for electricity, to the strong diurnal fluctuations of the dispersed emissions associated with road transportation (Nassar et al., 2013). This strong spatial and temporal patterning of the fossil fuel emissions interacts with the spatiotemporal variability of atmospheric transport, forming distinct patterns of the fossil fuel signal in atmo-

spheric $CO_2$ (Feng et al., 2016). Of particular relevance are the diurnal and the seasonal changes in emissions, since they tend to co-vary with atmospheric transport, which can lead to atmospheric $CO_2$ concentration gradients due to a rectification effect (Denning et al., 1995; Zhang et al., 2016). Such unaccounted for variations in the fossil fuel signal would bias the biospheric signal in atmospheric inversion frameworks, hindering us from developing a better understanding of the role of the

land biosphere as a carbon sink. At the same time, this strong temporal patterning of the emissions creates also distinct signals that might be used to detect or track the fossil fuel signal.

In fact, several studies already explored the possibilities to detect the fossil fuel signal (Ciais et al., 2014; Nassar et al., 2013). These include a range of methods and systems, including bottom up methods based on surface observation systems (Shiga et al., 2014; McKain et al., 2012; Keller

et al., 2016), CO and radiocarbon based methods (Levin and Karstens, 2007; Van Der Laan et al., 2010; Vogel et al., 2013), airborne measurements (Turnbull et al., 2011), satellite constraints (Kort et al., 2012), and top-down approaches on the basis of atmospheric inversions (Ogle et al., 2015; Lauvaux et al., 2016; Brioude et al., 2013). Spatially, the focus ranges from point scale emissions





(Bovensmann et al., 2010; Velazco et al., 2011; Turnbull et al., 2016), or urban-scale (Newman et al.,
2013; Bréon et al., 2015; Turnbull et al., 2015; Pillai et al., 2016) to regional and global (Keppel-
Aleks et al., 2013; Basu et al., 2016).

A necessity to successfully deploy any of these different detection approaches is a good under-
standing of the spatiotemporal dynamics of the fossil fuel signal over a scale that is sufficiently large
in order to avoid an unacceptably high sensitivity to the lateral boundary conditions, i.e., over scales
exceeding a few 100 km. A successful detection also requires a good understanding of the contribu-
tion of the other processes affecting atmospheric $CO_2$ variations, namely the exchange fluxes with
the land biosphere and with the ocean, respectively. Further, often it would be quite useful to know
the source processes responsible for the fossil-fuel $CO_2$ signature, i.e., what fraction of the signal
stems from emissions from a coal-fired power plant and what part from road transportation. This
helps, e.g., with the assessment of how the implementation of a particular policy affects the fossil
fuel signature, such as e.g., the shutting down of coal- fired power plants.

Few studies have taken a continental to global perspective on the fossil fuel signal (Keppel-Aleks
et al., 2013), as the focus in the last few years had been on urban areas (McKain et al., 2012; New-
man et al., 2013; Kort et al., 2012), or just whether the emissions in the city be detected or not (Hase
et al., 2015; Pillai et al., 2016). In addition, comparatively less work has been carried out in Europe
(Schneising et al., 2008), and the majority of those used relatively coarse resolution atmospheric
transport models, resulting in relatively washed-out gradients of the fossil fuel signal over Europe
(Keppel-Aleks et al., 2013; Peylin et al., 2011), or few of them focused on whether the potential re-
duced emissions could be discerned by current observation methods in this region or not(Levin et al.,
2011). Furthermore, little consideration has been given to the temporal variations of the emissions.

The main objective of this work is to fill these gaps, and to develop a quantitative understanding
of the fossil fuel $CO_2$ signal in Europe. To this end, we employ a forward modeling approach using a
high resolution atmospheric transport model for Europe, forced with finely resolved fossil fuel emis-
sion fluxes in time and space. In this paper, we will (i) investigate the magnitude of the contribution
of the fossil fuel $CO_2$ signal to the variations in total $CO_2$; (ii) understand how the high temporal
resolution considered in the fossil fuel emissions affect the fossil $CO_2$ signal; and (iii) determine the
detectability of a reduction of fossil fuel emissions from different sources through changes in the
column mean $CO_2$ as seen, e.g., by a satellite-based observing system. We first describe the model
and methods, followed by the evaluation of the model in the third part. We then present the results,
followed by a discussion of each of the aforementioned three main topics, and then conclude with a
summary and an outlook.





## 2 Methods and Data

To simulate the fossil-fuel $CO_2$ over central and southern Europe in the context of the variations in total atmospheric $CO_2$, we employ a regional high- resolution atmospheric transport model for

the European domain and prescribe lateral and boundary conditions for the various components that constitute atmospheric $CO_2$. These include the fossil fuel emissions, the $CO_2$ exchange fluxes with the land and ocean surfaces, and the lateral atmospheric $CO_2$ boundary conditions. The simulations cover the period April 2008 until April 2009. The following subsections describe the methods and data in more detail.

### 2.1 Atmospheric transport model

The simulations were undertaken with the limited-area atmospheric prediction model COSMO (Consortium for Small-scale Modeling) (Baldauf et al., 2011) Version 4.23. We employed the COSMO-7 setup developed by the Swiss Federal Office for Meteorology and Climatology (MeteoSwiss) for the purpose of providing boundary conditions for the inner COSMO-2 grid used for forecasting the

weather in Switzerland. The COSMO-7 setup has a grid spacing of 6.6 km and its domain covers central and southern Europe (35.16°N/9.80°E (lower left) to 56.84°N/23.02°E (upper right)) (see Fig. 1).

The COSMO model is based on the primitive hydro-thermodynamical equations describing compressible non-hydrostatic flow in a moist atmosphere without any scale approximations. The model

equations are solved numerically on a rotated latitude-longitude grid, with terrain-following coordinates in the vertical (60 vertical levels, and lowest level at 10 meters), using an Eulerian finite difference method. Parameterization schemes are used to resolve the sub-grid scale physical processes such as vertical diffusion (turbulence), convection, radiation, and soil processes. A tracer transport module was recently added to the COSMO model, permitting the online transport of passive trac-

ers in a manner that is fully consistent with the physics of the model (Roches and Fuhrer, 2012). In our setup, advective transport was accomplished with a 3-dimensional semi-Lagrangian scheme. The tracers are transported in the model as moist air mass mixing ratios $q_{CO2}$. Values reported here are provided as dry air mole fractions $\chi_{CO2}$, calculated as $\chi_{CO2} = q_{CO2}/(1 - q_{H2O})M_{dry}/M_{CO2}$, where $q_{H2O}$ is the specific humidity and $M_{dry}$ and $M_{CO2}$ are the molar masses of dry air and $CO_2$,

respectively. The dry air column average mixing ratio is calculated as $XCO_2 = (\sum_{k=1}^{K}(p(k+1/2) - p(k-1/2))q_{CO2}(k))/(\sum_{k=1}^{K}(p(k+1/2) - p(k-1/2))(1 - q_{H2O}(k))) * M_{dry}/M_{CO2}$. K is the vertical level, and p pressure, which is at the staggered level(Baldauf et al. (2011)is recommended for more information).





## 2.2 Fossil fuel emissions

The fossil fuel emissions for $CO_2$ were generated by merging a relatively coarse emission inventory
for the regions outside Switzerland (EDGAR v4.2_FT2010, approx. 10 km, (Janssens-Maenhout
et al., 2012)) with a high-resolution (0.5 km) emission inventory for Switzerland. The latter was pro-
duced by the company MeteoTest specifically for the CarboCount CH project. The annual emissions
from this merged product for the year 2008 amount to 2.54 Pg $CO_2$ over the domain, representing

about 10% of the global emissions of that year (Le Quéré et al., 2016). We merged the emission
categories from the two inventories to 5 large emission categories, i.e., power generation, residential
heating, road transportation, industrial processes, and others. Even though each of these different
categories have a distinct emission pattern, many of them co-occur in the large metropolitan areas,
leading to a very patchy emission pattern with strong emission hotspots, and extensive regions with

relatively low emission densities (Fig. 1).

These emission inventories are given for each emission category as annual totals for each grid
cell, requiring us to multiply them with time functions to generate hourly timeseries of the fossil fuel
emissions at each location (Nassar et al., 2013). The time functions we employed were originally
generated by the University of Stuttgart (Institute für Energiewirtschaft und Rationelle Energiean-

wendung, IER) for the GENEMIS project (Friedrich and Reis, 2004) and have been used since in
several air quality modeling studies. The time functions are comprised of diurnal, weekly and sea-
sonal components and are specific to each of the main economic sectors (activities collected in the
Selected Nomenclature for Air Pollution (SNAP) codes) (Kuenen et al., 2014). The time functions
(except for the daily one) vary also by country, and are locally adjusted to reflect local time. Some

reassignments were necessary to align the categories used in EDGAR v4.2 and the CarboCount CH
inventory (both following IPCC guidelines) with the SNAP categories as described in the supple-
mentary material.

The time functions differ greatly between the various categories, reflecting their very different
time course of activities over the average day, week or year (see Fig. 2a,b). Among all diurnal time

functions, road transportation has the largest diurnal variability and is characterized by two peaks
during the day reflecting the rush hour periods (local time 8:00-9:00 and 17:00-18:00). Also residen-
tial/commercial combustion has a distinct diurnal cycle with two peaks. In contrast, the emissions
from industrial processes and fossil- fuel fired power plants vary less over the course of the day and
also have only one peak. The time functions for the day-of-week reflect primarily the lower industrial

activities and traffic during the week end, while most other sectors continue to emit at only slightly
smaller rates (see Fig. 2a). Combining all the sectors together, emissions during the weekend are
15-20% lower than during the week. The seasonal time functions depend primarily on the local cli-
matic conditions (see Fig. 2b), with northern and central European countries having a maximum in
winter due to their heating requirement, while there is little seasonality in emissions in the southern

European countries.





In order to be able to trace the fossil fuel signature in atmospheric $CO_2$ back to the emitters, we consider separate fossil-fuel tracers for ten different countries (or groups of countries) in our atmospheric transport model (see Fig. 1). Each of these tracers receives only the emissions from its respective country or group of countries, while elsewhere, the emissions are set to zero. Due to the linearity of atmospheric transport and the absence of any transformation of $CO_2$ in the atmosphere, the individual country-based tracers can then be summed to obtain the total fossil fuel $CO_2$ signal. In addition, in order to determine the contribution of the different $CO_2$ emission categories to the total fossil fuel $CO_2$, we also included five additional fossil fuel tracers, one each for the five categories we consider, i.e., power generation, residential heating, road transportation, industrial processes, and others. For these 5 tracers, we used time-invariant emissions, permitting us to assess also the role of the time variations in emissions on the fossil fuel $CO_2$ signal. In total, we included 17 fossil fuel tracers (10 countries, 5 sectors, and total fossil fuel $CO_2$ with time varying emission, and total fossil fuel $CO_2$ with time constant emission) in our high-resolution simulation study.

### 2.3 Other $CO_2$ component fluxes

In order to simulate the distribution of total atmospheric $CO_2$, we also include in our model three other $CO_2$ components, namely background $CO_2$, the terrestrial biospheric $CO_2$ and the oceanic $CO_2$ components. The background $CO_2$ represents that part of the atmospheric $CO_2$ that enters the domain through its boundaries. These boundary concentrations are provided by the post- assimilation results of CarbonTracker Europe (Peters et al., 2010). For the terrestrial biospheric $CO_2$ component, we used the hourly terrestrial biospheric fluxes from the Vegetation Photosynthesis and Respiration Model (VPRM) (Mahadevan et al., 2008). For the oceanic $CO_2$ component, we combined the monthly air-sea $CO_2$ flux estimates for the Atlantic from Landschützer et al. (2013) with the annual mean flux estimates for the Mediterranean by D'Ortenzio et al. (2008). As the oceanic flux contribution is small, no attempt was made to add higher frequency variability.

### 2.4 Simulations

The hindcast simulation started on March 1, 2008, with the initial and boundary conditions for meteorology taken from the operational hourly COSMO-7 analyses of MeteoSwiss and the initial and boundary conditions for atmospheric $CO_2$ provided by CarbonTracker Europe (Peters et al., 2010). The model was then run for 13 months until April 30, 2009. No assimilation of any meteorological data was performed. The lateral and boundary conditions for the total of 18 $CO_2$ tracers considered (15 fossil fuel, 3 other components) were prescribed as described above. We consider the first month as a spinup, and use the subsequent 12 months for our analyses.





## 3 Evaluation

### 3.1 Total atmospheric $CO_2$

We evaluate our COSMO-based results for the total atmospheric $CO_2$ concentration (computed by summing the fossil fuel component with the three others) by comparing them to the measurements from four sites in central Europe, namely Mace Head (MHD, 3.33°N, 9.90°W, 5m above ground, coastal site, 15 m a.s.l.), Cabauw (CBW, 51.97°N, 4.92°E, 20m, 60m, 200m above ground, flatland, near urban site, 0 m a.s.l.), Hegyhatsal (HUN, 6.95°N, 16.65°E, 10m, 48m, and 115m above ground,

continental site, 248 m a.s.l.) (Geels et al., 2007), Puy de Dome (PUY, 45.46°N, 2.58°E, mountain site, 1480 m a.s.l.). In order to minimize the impact of local influences, we use the average $CO_2$ concentrations between 12:00 and 18:00 local time, i.e., the time of day of maximum vertical mixing.

The modeled atmospheric $CO_2$ records at these four sites agree well with the observed ones (see Table 1). The correlation between the modeled and observed values exceed 0.7 at all sites and

240 heights. The highest correlation is found at Mace Head (MHD) (>0.81). This is due to the relatively steady conditions that characterize this relatively clean coastal site. Influence from air pollution is only observed during episodes of transport from the United Kingdom and continental Europe, which are very well captured by the model. The correlations are somewhat lower at the more polluted and more continental sites, i.e., between 0.72 and 0.78 at the coastal tall tower station Cabauw (CBW) in

the Netherlands, and around 0.8 at the continental tall tower station Hegyhatsal (HUN) in Hungary. Even the atmospheric $CO_2$ variations at the mountain top site Puy de Dome in France are well captured (r = 0.75).

COSMO tends to systematically underestimate the observed $CO_2$ concentration at most of the stations and levels, except at the coast of Ireland (MHD), where it is overestimated by 0.3 ppm (Table

1). The biases tend to get larger with increasing continentality of the sites, and the associated higher complexity of the surrounding terrain and other influencing factors. At the Cabauw site (CBW), the biases amount to between -0.8 and -1.6 ppm, while in central Hungary (HUN) the biases are already more than -4 ppm at all vertical levels. In general, this may be related to COSMO-7 ventilating the planetary boundary layer too strongly, particularly in winter time under weakly stratified conditions.

This is especially acute for the HUN site, because the air in the lowest atmospheric levels tends to get trapped at this site owing to the winter-time prevalence of anticyclonic conditions in the Carpathian Basin (Haszpra et al., 2012). An alternative explanation is that the biospheric sink simulated by VPRM is too strong, as discussed later.

Even though COSMO exhibits some biases in the mean, it captures the observed variability gen-

260 erally well (Table 1). In particular, COSMO reproduces the strong gradient in variability between the coastal site Mace Head (∼6 ppm) and the continental site in central Hungary (∼12 ppm), reflecting primarily differing contributions of synoptic variations on atmospheric $CO_2$. However, the





absolute magnitude of the variations are not matched by our simulations, with COSMO consistently underestimating the observed variability.

Overall, the evaluation of the total atmospheric $CO_2$ concentration reveals a good agreement with the observations, both in terms of mean and variability. The low and positive bias at the Mace Head site, where the contribution of the background $CO_2$ component dominates, suggests that this component is overall well modeled and likely not responsible for the bias at the other sites. This bias is likely due to the superposition of biases in atmospheric transport (as argued for the HUN site) and

biases in the underlying boundary conditions for the fossil fuel emissions and/or terrestrial fluxes. Since the contribution of the oceanic fluxes is very small, this component can be excluded as an explanation. Unfortunately, we do not have observationally-based estimates of the fossil fuel or terrestrial biosphere components at the four sites discussed so far, requiring us to use data from other sites for further evaluation.

### 3.2    Fossil fuel $CO_2$ component

Estimates of the fossil fuel component in atmospheric $CO_2$ are available for our model simulation period from Lutjewad in the Netherlands (LUT, 6.35° E, 53.4° N, 1 m a.s.l.) (Van Der Laan et al., 2010; Bozhinova et al., 2014) and from Heidelberg (HEI, 49.417°N, 8.675°E, 116 m a.s.l.) (Levin and Karstens, 2007). Both estimates are based on a combination of concurrent CO and $^{14}CO_2$ mea-

surements and represent the fossil fuel induced offset relative to a regional background. They are thus comparable to our modeled fossil fuel component, as this reflects the offset relative to the domain-wide background induced by the lateral boundary conditions. Lutjewad is located on the Waddensea dike in the north of the Netherlands, influenced by the highly populated and industrialized areas in the Netherlands and in northwestern Germany (Ruhr area). The Heidelberg station is located near an

urban center with considerable fossil fuel emissions.

At the Dutch site LUT, the daily average fossil fuel $CO_2$ component simulated by our model compares well with the observations (r =0.73, mean bias -4 ppm) (see Fig. 3a). Generally, the model reproduces the observed variability, especially in summer, when the fossil fuel $CO_2$ component is low owing to the deep mixing in the atmosphere. But the model underestimates the estimated fossil

fuel $CO_2$ component substantially in winter. This may be due to several reasons. First, the model may transport signals too quickly out of the planetary boundary layer, which is a known problem of many atmospheric transport models under stratified conditions typical of wintertime (see also above) (Holtslag et al., 2013). Second, our wintertime emission inventory in the region might be too small, owing to, for example, our underestimating the strength of the seasonal signal in the

time functions. Third, the observations might be biased high. One reason is that these reconstruction rely on a constant ratio between CO and $^{14}CO_2$, which may lead to an underestimation of the $^{14}$C-CO ratio compared to real values at some time of the year, and subsequently overestimation of the inferred fossil fuel $CO_2$ (Van Der Laan et al., 2010; Bozhinova et al., 2014).





At Heidelberg, our model captures the fossil fuel $CO_2$ component even better, particularly since
the model has a very small mean bias of 0.75 ppm. Also the day- to-day and the seasonal variations
are well represented with a correlation coefficient of 0.72. The model's (small) overestimation of
the fossil fuel component may be due to our prescribing all emissions at the surface, while the
fossil-fuel fired power plants that contribute substantially to the fossil fuel $CO_2$ at this site tend to
have an effective emission height quite some distance above the ground due to the height of the
stacks and the additional rise of the buoyant plumes (Vogel et al., 2013). Another reason might be
an overestimation of the emissions in our emission inventory EDGAR - an explanation furthered
by EDGAR's emission being higher than those of IER (Peylin et al., 2011). Especially assuring,
and particularly so in comparison to the situation at LUT, is the COSMO model's ability at HEI to
capture most of the variability and amplitude of the fossil fuel component in winter. An exception
are the observations from late December and early January, where the data include a number of
exceptionally high peaks. These peaks may be the result of very strong local trapping of the emitted
fossil fuel $CO_2$ by e.g., a local inversion situation, i.e., a process that our model cannot properly
resolve.

Despite these discrepancies, the good to excellent evaluation results provide us with good con-
fidence to use our COSMO-7 based system to investigate the spatio- temporal variability of the
fossil-fuel $CO_2$ in central and southern Europe. It is particularly encouraging to note the good agree-
ment not only for the fossil fuel $CO_2$ component, but also for total atmospheric $CO_2$. The presence
of an overall negative bias in the total atmospheric $CO_2$ in the absence of such a bias in the fossil
fuel component suggests that the bias comes from the terrestrial biospheric component. This could
be due to our VPRM-based estimates of the net fluxes being too negative as suggested by Oney
et al. (2016, in review), i.e., suggesting a too strong sink for central and southern Europe, or for our
model simulating a too small diurnal and/or seasonal rectification effect (Denning et al., 1995), i.e.,
a too small correlation between the time variations in the terrestrial exchange fluxes and atmospheric
transport/mixing. This deficiency does not impact our results much, since our focus will be on the
spatio-temporal variability of the fossil fuel $CO_2$ signal.

## 4 The spatiotemporal pattern of the fossil fuel $CO_2$

### 4.1 The spatial pattern

In the annual mean, computed from data from all times of the day, the fossil fuel component of
atmospheric $CO_2$ in the surface layer ($\sim$10 m above ground) amounts to more than 10 ppm across
wide swaths of central Europe (Fig. 4a). In large metropolitan areas, such as in western Germany
(Ruhrgebiet), Berlin, London, Paris, and Milan, the annual mean fossil fuel component exceeds
even 30 ppm. To first order, the distribution of the surface fossil fuel $CO_2$ reflects the distribution
of the emissions (see Fig. 1), suggesting a somewhat limited efficiency of atmospheric transport


and mixing to disperse the signal laterally. In mountainous regions this is clearly a consequence of
335 topographic constraints; elsewhere this is largely a result of the strong spatial gradients in emissions,
which remain conserved in the annual mean due to the overall diffusive nature of the dispersion.
Nevertheless, a substantial amount of the emitted $CO_2$ is being transported away, leading to a size-
able fossil fuel $CO_2$ signal extending far into the oceans surrounding Europe, especially the North
Sea.

Despite this lateral transport, the relatively good conservation of the spatial gradients in emissions
sets our results distinctly apart from previous studies, where the fossil fuel $CO_2$ signal was modeled
to be very smooth in space and on average also substantially smaller. For example, compared to the
results obtained with the medium-resolution ($0.5°$) Regional Model (REMO) (Peylin et al., 2011),
one can detect in our simulations nearly all major metropolitan regions and other fine-scale features,
such as individual fossil-fuel fired power plants (e.g., in eastern Germany). This is primarily the
result of the high horizontal and vertical resolution of COSMO permitting this model to conserve
the spatial gradients well. This good conservation is particularly well illustrated when considering
snapshot distributions of the fossil fuel $CO_2$ for individual seasons (Fig. 5). This figure also shows
the strong impact of the transport and dilution by the diurnal variations of the planetary boundary
layer, whose impact is particularly strong in summer.

For much of Europe, the fossil fuel component is the dominant contributor to the spatial gradients
in annual mean atmospheric $CO_2$ (Fig. 4b-d). In many places it accounts for nearly all of the spatial
gradients, with the contribution of the background and the terrestrial biospheric component being
substantially smaller. The latter shows gradients up to 10 ppm (Fig. 4c), while the background signal
does not exceed a few ppm (Fig. 4d). In the big cities, the fossil fuel $CO_2$ component represents
even a sizeable fraction (10%) of the total $CO_2$ concentration. This dominance of the fossil fuel
component together with its highly patterned nature owing to the many point sources leads to a
hotspot pattern in the near surface map of total atmospheric $CO_2$ over much of Europe (Fig. 4b).
However, due to lower emissions in southwestern Europe, the fossil fuel $CO_2$ signal is less strikingly
visible there compared to central Europe, while the biospheric signal is stronger. This results in a
relatively uniform spatial pattern of atmospheric $CO_2$ across Europe (Fig. 4b). Also the relatively
low $CO_2$ concentrations in the mountain regions, such as the Alps, Apennines, Pyrenees and central
France, reflect the much lower contribution from the fossil fuel component.

Naturally, when investigating the column averaged dry air mole fractions ($XCO_2$), i.e., the prop-
365 erty typically measured by remote sensing from a satellite, the annual mean gradients of the fossil
fuel component are much smaller than those seen at the surface (see Fig. 6a). This is a consequence
of the lateral gradients being much weaker aloft, owing to a more effective transport and mixing.
As a result, most of the hotspot nature seen in the surface concentration pattern is blurred in $XCO_2$.
Also the magnitude of the signal is much weaker. While the surface signal of the fossil fuel $CO_2$
signal amounted to more than 30 ppm in strong emissions regions, the signal in the column aver-



aged annual mean $X\mathrm{CO_2}$ hardly exceeds 2 ppm. The impact of the predominant westerly air-flow becomes much more obvious in the column averaged dry air mole fraction $X\mathrm{CO_2}$, with the fossil fuel component revealing a clear eastward increase that is substantially stronger than the gradient in the underlying emissions.

The relative dominance of the fossil fuel component over the other components of atmospheric $\mathrm{CO_2}$ is much weaker when considering the column averaged dry air mole fraction of $\mathrm{CO_2}$ (see Fig. 6b-d). As a result, the total $X\mathrm{CO_2}$ is made up of three relatively equally sized contributions, with the fossil fuel $\mathrm{CO_2}$ signal continuing to dominate the $X\mathrm{CO_2}$ variations in the major metropolitan areas. Contrary to the annual surface pattern, where $\mathrm{CO_2}$ tends to increase eastward, the highest

$X\mathrm{CO_2}$ are found in southwestern Europe with a trend toward lower values going eastward. This is partly a consequence of the lateral boundary conditions for atmospheric $\mathrm{CO_2}$, which tend to lead to the advection of high background $\mathrm{CO_2}$ into the domain from the southwest. But the most important reason is the strong negative terrestrial biosphere signal over Europe, reflecting the sizeable carbon sink in European forests in the last decade (Reuter et al., 2016, in press). Interestingly, the relatively

uniform negative distribution for $X\mathrm{CO_2}$ contrasts with a more patterned biospheric signal in the lowest atmosphere (Figure 4c), where the strong negative signal is restricted to central Europe, while much of southern Europe has a positive annual mean biospheric signal. The likely reason for this difference is the biospheric rectification effect (Denning et al., 1995), which tends to lead a vertical redistribution of $\mathrm{CO_2}$, i.e. positive values in the lower atmosphere and negative ones aloft.

In most of Europe, this rectification signal is relatively small in comparison to the annual mean biospheric component, so that the latter determines the overall signal. But in southern Europe, where the biospheric fluxes tend to be smaller in magnitude and in the annual mean to be near zero, the rectifier effect can dominate, explaining the positive signals in the surface layer (Figure 4c) and simultaneously the negative signals when the biospheric signal is integrated vertically (Figure 6c).

## 4.2   The temporal variability

The temporal variability of the fossil fuel $\mathrm{CO_2}$ signal at the surface is very large, leading to a standard deviation around the annual mean of 30 ppm or more in the hotspot regions (Figure 7a). These hotspots correspond largely to the regions of highest emissions (Figure 1). But this high variability is not only a result of the temporal variability of the emissions, but arises also from the interaction of

variability in atmospheric transport and mixing with the strong lateral gradients seen in the snapshot figures (see Fig. 5).

    A similar pattern of variability is seen in surface atmospheric $\mathrm{CO_2}$ (Figure 7b), suggesting that the fossil fuel $\mathrm{CO_2}$ is a major determinant not only of the annual mean spatial distribution of atmospheric $\mathrm{CO_2}$, but also of its temporal variability. This is confirmed by Figure 8a, which shows the relative

contribution of the fossil fuel $\mathrm{CO_2}$ signal to the temporal standard deviations of atmospheric $\mathrm{CO_2}$. In many places, particularly in Europe's major metropolitan areas, but also in many urban areas



across Europe, the fossil fuel signal dominates the variability in atmospheric $CO_2$. But the high fossil fuel contribution is not limited to the urban areas. In fact, the region delineated by having a 50% contribution or more extends over much of northern central Europe, including the North Sea

(see Fig. 8a).

In order to better understand the origin of the strong variability, we decomposed the variability into seasonal, synoptic and diurnal contributions. The seasonal variation component was derived by averaging the data on a monthly basis and by subtracting the annual mean. The synoptic component was then computed by subtracting from the data the time series of the monthly means and then

forming daily averages of these deseasonalized data. Finally, the diurnal variability was derived by subtracting the seasonal and synoptic components from the data.

This decomposition reveals that the contribution of the fossil fuel $CO_2$ to the total variability of atmospheric $CO_2$ varies greatly depending on the temporal scale considered (Figure 8). While the fossil fuel contribution is comparably small on seasonal timescales (Figure 8b), the contribution on

synoptic and particularly on diurnal timescales is actually very large, exceeding 60% across nearly the entire northern part of central Europe (Figure 8c-d). The small contribution on the seasonal timescales is the result of the dominance of the seasonal cycle of the biospheric fluxes in most of Europe. An exception are a few places in northern Europe and in the very south of our European domain. We interpret this to be caused primarily by the relatively strong seasonality of the fossil fuel

emissions in these regions, owing to the strong summer-time requirement for cooling in the south and the strong winter-time demand for heating in the north.

The pattern of the fossil fuel contribution on synoptic timescales is very similar to that of the total contribution, meaning its contribution is one of the dominant contributions to the total temporal variability. This is consistent with synoptic variations also being among the strongest contributors to

atmospheric variability, owing to baroclinic waves and frontal systems being formed out of the strong baroclinicity that characterize the mid-latitudes. These synoptic weather events transport the emitted $CO_2$ also quite efficiently outside the main metropolitan areas, explaining the widespread signal of the fossil fuel contribution to the total variance of atmospheric $CO_2$. Even larger than the fossil contribution to synoptic variability is the contribution on the diurnal timescale, with the fossil fuel

$CO_2$ contributing more than half of the variability over most of Europe. This high variability comes from the interaction of the diurnal variability of the fossil fuel emissions, with the strong diurnal variability of atmospheric transport, particularly the diurnal mixing of the planetary boundary layer. This co-variability between fossil fuel emissions and atmospheric transport exceeds that between the biospheric fluxes and atmospheric transport over the entire year, owing to the latter fluxes being large

and relevant only during the spring/summer period, while the fossil fuel emissions are relatively high during most of the months of the year, particularly close to the sources.



## 5 Discussion

The analyses of the results raise a number of questions that we would like to discuss next. First, why
is the diurnal variability so high, and in particular, what is the contribution of our consideration of
diurnal (and seasonal) variations in $CO_2$ emissions on the simulated fossil fuel $CO_2$ signal? Further,
is there an impact beyond the variability, e.g., on the mean fossil fuel $CO_2$ signal? Second, what is
the contribution of the various sectors on the fossil fuel $CO_2$ signal and in what way do emissions
from one country influence the fossil fuel $CO_2$ signal in another country? Third, how can we use
the insights gained from the study of the fossil fuel $CO_2$ signal to develop optimal strategies for
detecting changes in fossil fuel $CO_2$ emissions? We discuss each of these three questions next.

### 5.1   The impact of variations in fossil fuel emissions on atmospheric $CO_2$

In order to elucidate the role of the temporal variations in fossil fuel emissions on the fossil fuel $CO_2$,
we contrast the results of our standard simulation with time-varying emissions with those where the
fossil fuel emissions were kept constant over time. The annual emissions are identical for the two
cases, but the time constant case has, on average, considerably higher emissions in summer and at
night.

The contrast between these two cases shows only a small change in the high diurnal variability of
atmospheric $CO_2$ seen in Figure 8d, implying that the contribution of the diurnal variations in fossil
fuel emissions is less important than other factors (results not shown). The largest contributions
are found around some of the large metropolitan areas (e.g., London, Paris, Milan), but they do
not exceed 10%. Thus the majority of the diurnal variability in the fossil fuel $CO_2$ stems from the
diurnal variations in atmospheric transport and mixing acting on the strong horizontal gradients in
emissions.

While not contributing much to the diurnal variability in the fossil fuel $CO_2$, the consideration of
the time-varying emission matters quite substantially for the annual mean distribution of the fossil
$CO_2$ signal. Figure Fig. 9a reveals that the annual mean fossil $CO_2$ signal in the simulation with
time varying emissions is substantially lower over wide swaths of Spain, Italy, the Benelux countries,
(western) Germany and the UK compared to the simulation where fossil fuel emissions were kept
constant. The strongest negative signals are found close to the strongest emitters in these countries,
with magnitudes exceeding several ppm. But the magnitude of the signal does not correspond to the
magnitude of emissions, since regions with comparably low emissions such as Spain, have signals
that are as large as those in high emission regions of the Netherlands. The relatively large signals in
southern Europe are likely due to the stronger PBL dynamics in these regions throughout the year in
comparison to central and northern Europe. Some regions also have a positive signal from the time-
varying emissions, such as parts of France and northeastern Germany. Thus the interaction between
the variations in fossil fuel emissions and the variations in atmospheric transport and mixing leads

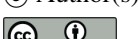



to a substantial net signal in atmospheric $CO_2$, even though the total emissions in both cases are identical.

This net signal represents a fossil fuel-driven rectification effect (Zhang et al., 2016) in analogy to the rectification effect associated with the terrestrial biosphere (Denning et al., 1996; Larson and Volkmer, 2008), i.e., a signal that is due to the co-variance of emissions and atmospheric transport/mixing. Its (mostly) negative sign emerges from the fact that when the emissions are large, e.g., during the day, the transport and mixing away from the surface is strong, diluting the fossil fuel signal in atmospheric $CO_2$. In contrast, when the emissions are small, e.g., during the night, the transport and mixing tends to be weak. Taken together, this results in a more efficient dilution of the emissions in the time-varying emission case compared to the time-invariant case, thus explaining the mostly negative sign of the fossil fuel rectification effect.

This explanation is supported by the mostly positive correlation between the height of the planetary boundary layer (PBL) and the fossil fuel emissions, since the height of the PBL is a direct measure of the magnitude of the dilution in the lowest levels of the atmosphere (Figure Fig. 9b).

But there are a number of notable exceptions. For example, wide swaths of northeastern Germany and Poland and some places in central France have a positive rectification signal. Further, there are places where the co-variation of fossil fuel emission and the PBL is negative, yet the fossil-fuel rectification effect is still negative (e.g. the Ruhr valley region in western Germany), suggesting that our explanation does not cover all aspects. In response, one first needs to recognize that not only PBL but also other temporally varying phenomena, such as local atmospheric circulation patterns (e.g. mountain winds, sea-breezes) can lead to co-variability between emissions and transport/mixing, creating a rectification signal that can differ in sign. The contribution of the sea-breeze can be identified quite clearly by the strong negative sign along most of the coastline between southern Europe and the Mediterranean. Second, the local timing between the growth and decay of the PBL and the emissions can be quite different, owing in part, to the substantially different time functions for the different emission categories and their different local contributions (Figure 1). For example, in regions with a large contribution from road transportation, the local emissions have a strong peak in the early morning hours, when the PBL is still shallow, leading to a high signal there, while emissions are lower when the PBL is at its maximum in the early afternoon. This would create a positive rectification signal. Finally, in certain places, also the seasonal rectification appears to play a role, i.e., the seasonal co-variations of the emissions with the PBL height. In fact, in many places the magnitude of the correlation between emission and PBL height on seasonal timescales exceeds that on diurnal timescales. This seasonal variation is particularly large for residential heating, which is maximum in winter when the PBL is low, leading to a positive seasonal rectification. This effect likely contributes to the negative correlations between emissions and PBL height in large urban centers such as Paris (Figure Fig. 9b). We suspect that such seasonal effects are also the primary reason for the positive rectification signal in northeastern Germany and northern Poland. In southern Europe, these seasonal





co- variations tend to lead to a negative fossil-fuel rectification effect, since the emissions peak in
summer (Figure 2b), when the PBL height is at its seasonal maximum.

The magnitude of the fossil fuel rectification effect is smaller than the rectifier effect induced by
the exchange fluxes with the terrestrial biosphere (Zhang et al., 2016), but still quite substantial.
Thus, the fossil fuel rectification effect clearly needs to be taken into consideration when model-
ing the atmospheric fossil fuel $CO_2$ signal, highlighting the need to use and apply accurate time
functions. Our results thus clearly support the results of Nassar et al. (2013), who demonstrated the
substantial impact of the consideration of time-varying emissions on atmospheric $CO_2$. We extend
their result by demonstrating an effect on the annual mean fossil fuel $CO_2$, suggesting that special
attention needs to be given to the relative timing of variations in atmospheric transport and mixing
and fossil fuel emissions. Our results confirm the recent findings by Zhang et al. (2016) who demon-
strated the fossil fuel rectification effect for the first time in a global model. Their signal is locally
smaller than ours, owing to their using a much coarser resolution model, but they also show that the
sign of the fossil fuel rectification effect tends to vary between timescales, with the diurnal being pri-
marily negative, while the seasonal rectification effect being positive. This supports our explanation
for the positive signals in northeastern Germany and northern Poland.

### 5.2   Fossil fuel $CO_2$ signal from different sources

Near the surface, the fossil fuel emissions from a particular region create a distribution that stays
mostly within the region of origin (see Fig. 10 a,b). The fossil fuel $CO_2$ is highly concentrated near
the localized areas of high emissions and then drops off quickly by distance with an e-folding spatial
scale of a few hundred kilometers. As a result, the fossil fuel signal tends to be relatively small
outside the region of origin, rarely exceeding 1 ppm in contrast to the $> 20$ ppm signal close to the
sources. The different magnitudes of the fossil fuel $CO_2$ signals from different regions largely reflects
the total emissions, but also the emission intensity, i.e., the emission per unit area. For example, with
a total emission of 0.59 Pg $CO_2$ yr$^{-1}$, Germany is the biggest source of fossil fuel $CO_2$ within
Europe, nearly double that of the second biggest emitter, i.e., France, yet Germany is almost half the
size of France, resulting in a considerably higher emission intensity over Germany.

A different picture emerges when considering $X CO_2$, i.e., the column averaged dry air mole
fraction $CO_2$. After having been transported aloft, where the fossil fuel signal can be much more
readily dispersed, the imprint of the emissions of any particular region to the fossil fuel $CO_2$ within
another region is actually quite large (Figure 10 c,d). In a small country, such as Switzerland, only
20% of the fossil fuel signature in $XCO_2$ above its territory stems from emissions within, while the
contribution of Germany alone is 21% and that of France 18% (Figure 10). A similar distribution of
sources is found for other small countries, such as Austria. In contrast, the fraction of the territorial
emissions to the total fossil fuel signal is quite a bit larger for large countries/regions, such as France
or Germany. In the latter case, more than 50% of its total fossil fuel $CO_2$ signal stems from emissions



within, with 4 countries contributing most of the remainder. The countries/regions with high overall emissions contribute, of course, also most strongly to the fossil fuel $CO_2$ signal in other countries, with Germany contributing 18% to the signal in France, 11% to that in Italy and 20% to that in the Netherlands. Owing to its lower total emissions, France just contributes 9% to the signal in Germany and 8% to that in Italy. Thus, as is the case with classical air pollution, the fossil fuel $CO_2$ does not stop at the national borders, but extends to continental scales.

Among all the processes, the $CO_2$ emissions from power plants dominate the fossil fuel distribution, with concentrations reaching up to 16 ppm in the northern part of the domain (see Fig. 12). The point-source nature of this emission sector is clearly visible in the surface distribution, as is the spatially distinct distribution owing to the large differences in power production in the different countries of central Europe. While France has very few fossil-fuel fired power plants as a result of its high reliance on nuclear and hydroelectric power plants, Germany, Italy, the Netherlands and Poland rely strongly on coal- and gas-fired power plants for their electricity production. This leads to a highly heterogeneous fossil fuel $CO_2$ signals of the power plant sector. In total, this sector contributes 31.8% to the total fossil fuel $CO_2$ signal in central Europe, which is slightly smaller than its contribution to emissions (32.8%). This small difference emerges from the somewhat stronger loss of the signal across the lateral boundaries from this sector relative to the signal from the other sectors.

The second largest fossil fuel $CO_2$ signal is generated by the emissions from the road transportation sector (22.0%) (Fig. 12d), with this share actually being somewhat larger than its share in total emissions (21.1%). The transportation sector signal is very smooth, owing to the distributed nature of the emissions from this sector (see also Fig. 1).

The $CO_2$ signal from the industrial and residential sectors are more granular than that from the transportation sector, but still not as distinct as the power plant sector, as there are less country specific policies impacting the $CO_2$ emissions from these sectors. The emissions and consequently the $CO_2$ signal largely follow population density. The residential sector (mostly heating) contributes 18.1% to the total fossil fuel signal in atmospheric $CO_2$, slightly larger than the emissions from the industrial sector (17.4%). These two shares in the signal very nearly reflect their shares in total emissions. The emissions from the 'other 'sectors (e.g., shipping, waste incineration, etc) is smaller, in comparison (10.7%), but not negligible.

The relative contribution of the emissions from the different sectors to the fossil fuel $CO_2$ vary strongly by region (Fig. 13). Clearly, close to major fossil-fuel fired power plants, this sector dominates, but elsewhere, any of the four major sources can take the leading role. For example, in Switzerland, Paris, and London, the emissions from the residential sector dominate the signal, while over much of southern and western Europe, the transportation sector dominates. The industrial sector dominates the signal in a few hotspot areas, where its emissions are high, but where there is no major fossil fuel fired power plants nearby.



These high spatial variations in the relative contribution puts the findings of Vogel et al. (2013) into a spatial context, as they reported for the Heidelberg site a dominance for emissions from power plants (28%), while the transportation sector contributed only 15%. This is a typical value for much of western Germany, reflecting the relative contribution of the different emission sectors (see also Fig. 1). But the contributions are very different, for example, for the CarbCount CH sites in Switzerland (Oney et al., 2015). At Beromünster, the transportation sector dominates over the other sectors, with nearly 70% stemming from this sector alone, while the contribution from power plant emissions is very low at this site, since Switzerland does not operate any fossil fuel power plants.

These large differences in the relative contribution from the different emission sectors have major implications for the analysis of the fossil fuel $CO_2$ and how it may change in response to mitigation measures. For example, these large differences will lead to substantial spatial gradients in the CO to $CO_2$ ratio in the fossil fuel signal, as the different emission sectors have very different CO to $CO_2$ emission ratios. Since CO is often used to identify the fossil fuel component from atmospheric $CO_2$ observations, these variations need to be carefully disentangled in order to properly diagnose the fossil fuel component. The strong variations in the contributions from the different sectors thus adds a substantial amount of uncertainty to the CO method (Oney et al., 2016, in review; Vardag et al., 2015). A second consequence concerns the detection of changes in emissions from the different sectors. Thus, with the transportation sector contributing little to the very large fossil fuel signal in much of the northeastern part of our domain, reductions in this sector will be difficult to discern in that region. In contrast, the high relative contribution of the transportation sector to the total signal in southwestern Europe makes it actually quite feasible to detect mitigation measures in this sector in that part of Europe, even though the overall signal might not be that high.

An important caveat of our simulations is the fact that the effective height of the emissions above surface was not considered, but rather all $CO_2$ was released into the lowest model level. As a consequence, the surface $CO_2$ signals from elevated stack emissions from power plants and residential heating are likely biased high relative to those from the transportation sector. Given the large contribution from power plant emissions, it will be important to accurately consider the effective emission height (including plume rise) in future simulations, a point that was also raised by Vogel et al. (2013).

### 5.3 The response of atmospheric $CO_2$ to an emission reduction

According to their intended nationally determined contributions filed with the United Nations Framework Convention on Climate Change (UNFCCC) in late 2015, the European Union and its member states have agreed to a binding target of an at least 40% domestic reduction in greenhouse gases emissions by 2030 compared to 1990 (http://www4.unfccc.int/Submissions/INDC/Published%20Documents/ Latvia/1/LV-03-06-EU%20INDC.pdf). A major question driving international policy making is to what degree such a reduction can be verified through independent means, such as through the monitoring of atmospheric $CO_2$ (Ciais et al., 2014, 2015). To address this question, we conducted several





sensitivity experiments to investigate how various reductions in the magnitude and types of emissions affect not only the annual mean fossil fuel $CO_2$ signal, but also its variability. The goal is to determine whether reduced fossil fuel emissions might be detectable by current and future observing systems, especially satellites.

Since $CO_2$ is a conservative tracer in the atmosphere at the time scales considered here, a uniform reduction in the emissions leads to a uniform and directly proportional reduction of its current distribution, i.e., a 30% reduction of total fossil fuel emission would simply lead to a 30% reduction of the fossil fuel $CO_2$ signal at the surface (Fig. 4a) and throughout the atmospheric column (Fig. 6a). Concretely, the fossil fuel $CO_2$ would be reduced by more than 4 ppm near the surface for vast stretches of central and northern Europe, with maximum reductions of 10 ppm or more in the emission hotspots (Figure 14a). This contrasts with the reduction in the averaged column annual mean $XCO_2$ amounting to just over 0.2 ppm in the regions where the surface decreases by 4 ppm or more (Figure 14b). A reduction of 0.5 ppm is reached in just a few isolated locations, generally characterized by a high density of point sources, primarily fossil-fuel fired power plants. Thus, given current measurement accuracies of better than 0.1 ppm for a ground-based observing network (Zellweger et al., 2016), a 30% reduction in the fossil fuel emissions is fundamentally easily detectable for such a system, although one needs to bear in mind the non-trivial task to separate the signal from the background variability. In contrast, such a reduction in the fossil fuel emissions is not trivial to detect by satellite observations for most regions (except around the big power plants) as it is very challenging to obtain and maintain accuracies better than 0.5 ppm by current space- based observing systems (Buchwitz et al., 2015). Furthermore, such high accuracies are only achieved when the data are averaged over large scales, i.e., order of 1000 km or more. Nevertheless, taking 0.5 ppm as the threshold for detection within a single pixel, a 30% reduction in fossil fuel emissions thus appears to be beyond the detectability, except for a few hotspot regions(Figure 14b). Even a 50% reduction would not be trivial to detect for a satellite-based system on the basis of changes in the column averaged dry air mole fraction .

Given these challenges, a potentially attractive second avenue for determining changes in fossil fuel emissions is the reduction in temporal variability of atmospheric $CO_2$ that goes alongside the reduction in the mean signal. This is particularly promising given the very high contribution of the fossil fuel $CO_2$ signal to the variability in atmospheric $CO_2$ (see Fig. 8). As is the case for the mean, the conservative nature of atmospheric $CO_2$ implies that a uniform reduction of the emissions will lead to a uniform and proportional reduction of the variability of the fossil fuel signal as well. However, this is not the case for the variability in total atmospheric $CO_2$, since co-variations between the fossil fuel signal and the signal from e.g., the terrestrial biosphere can lead to non-linear effects. For example, a negative correlation between the two components would lead to a situation where the variability of atmospheric $CO_2$ was smaller than that of the individual components. In such a case, a reduction of the fossil fuel emission would lead to a smaller decrease in variability than expected.



If the two components were positively correlated, the opposite would occur, i.e., the variability in atmospheric $CO_2$ would decrease more than expected.

Near the surface, the reduction in the temporal standard deviation and in the mean have nearly the same amplitude for most places (Figure 14c). This makes the analysis of changes in the temporal variability indeed an attractive option to enhance the detectability of changes in fossil fuel emissions.

This is much less the case for the annual mean $X CO_2$, where the standard deviation changes are in general much smaller than the changes in the mean, with just a few isolated places revealing changes in the standard deviation of 0.5 ppm or more that might be discerned by the current generation of satellites.

But in these isolated places, the analysis of the temporal variability might be an interesting option
even for satellite-based measurement systems (Fig. 15). In those places, indicated by the green circles in (Figure 14c), the changes in the temporal standard deviation are very large. Even for changes in emissions of around only 30%, the changes would be detectable for current satellites (Fig. 15). But the number of such sites is very low across Europe, making this not a general, but rather a specialized option.

The detection challenge is not simpler for other potential emission reduction scenarios, as outlined, for example in the EU roadmap (http://ec.europa.eu/clima/policies/strategies/2050/index_en.htm)). A 50% reduction in the emissions from power plants alone (representing a reduction of the overall emissions by 16%), results in the mean surface concentration of atmospheric $CO_2$ going down by more than 2 ppm over large parts of northwestern Europe, following the pattern of the
surface signal of this sector (cf. Fig. 12a). Alongside we find a substantial reduction of the standard deviation of surface atmospheric $CO_2$ by more than 2 ppm in these regions, with the hotspots of power plant emissions seeing a reduction in the standard deviation of atmospheric $CO_2$ of 5 ppm or more. The reduction of the average annual mean column $X CO_2$ is much smaller than that of atmospheric $CO_2$ at the surface, amounting to little more than 0.2 ppm over wide swaths of north-
ern Europe. The maximum reductions are of the order of 0.5 ppm in the proximity of large clusters of fossil-fuel fired power plants, i.e., generally too small to be detected. But, in these regions, the changes in the variability in $X CO_2$ is quite high, making this method again potentially attractive for detecting changes. In fact, in several regions, including some major cities, a 19% reduction of the fossil fuel emissions would result in a change of more than 0.5 ppm in the standard deviation, i.e.,
above detection level. This thus supports the findings of Pillai et al. (2016) that changes in fossil fuel emissions are fundamentally detectable over major cities or major point sources, but it also shows that this detection is very challenging.

The signals get even more difficult to discern if the emission reductions occur in individual sectors other than the power plants. For example, detectable signals by current generation satellites occur
only if industrial emissions are cut by more than 80% or if residential emissions are cut by more than 90%. Also country level emissions are not trivial to be clearly detected. A reduction in Germany by

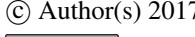



50% is potentially detectable by current satellites, with a maximum reduction of $X$CO$_2$ by 0.95 ppm. For most other countries, however, a 50% reduction in emissions is difficult to be detected.

All the analyses here relied on using the model output on all available days, i.e., we assumed perfect temporal coverage. This is overly optimistic, since cloud cover and other complicating factors (e.g., aerosol layers) will cause the coverage to decrease considerably, complicating the detection.

But regardless of this additional challenge, there is much additional information contained in high frequency observations of atmospheric CO$_2$. As we demonstrated above, the temporal variations are potentially highly useful for detecting fossil fuel emissions changes from various sources, especially those with a strong spatial granularity such as power plants or individual cities. For a routine monitoring of strong point sources, Velazco et al. (2011) therefore proposed a constellation of 5 satellites of type CarbonSat that combine imaging capability with a relatively wide swath (Bovensmann et al., 2010). Such a constellation would offer daily global coverage, though the presence of clouds would reduce the effective coverage considerably. As the precision and accuracy of satellite retrieved XCO$_2$ will improve in the future, that minimum change will go down as well.

## 6 Summary and conclusions

We have investigated the fossil fuel signal in atmospheric CO$_2$ over southern and central Europe using a regional high-resolution atmospheric model forced with temporally and spatially highly resolved variations in the fossil fuel emissions. The assessment of the modeled atmospheric CO$_2$ with in-situ measurements across multiple sites across Europe reveals good to excellent agreement on all timescales considered with biases of less than 1 ppm, with the exception of the tall tower site Hegyhatsal in central Hungary. The model is also able to capture the reconstructed fossil fuel component at two sites quite successfully. Although the model tends to underestimate the amplitude of the daily averaged fossil fuel CO$_2$ in winter, the simulation matches fossil fuel CO$_2$ from both sites very well most of the time, revealing the high quality of the transport model and reasonable time profiles of the fossil fuel emissions used as input.

Over much of Europe, the fossil fuel CO$_2$ is a dominant component of the spatial variability of atmospheric CO$_2$, particularly near the surface. In some places, it even contributes significantly to the total (including background) CO$_2$, up to 110% in large urban centers and power plant plumes. Also the contribution to the temporal variability is very substantial. Fossil fuel CO$_2$ makes a particularly large contribution at synoptic and diurnal time scales whereas the seasonal variability is dominated by biospheric activity. The influence is not only large over the hot spot regions of fossil fuel emissions, but also over large areas downstream. In case of diurnal variability, fossil fuel CO$_2$ is the dominant component over wide areas of northern and western Europe.

Temporal variability of the emissions has a non-negligible influence on annual mean fossil fuel CO$_2$ mole fractions near the surface, due to diurnal and seasonal rectifier effects. Differences be-





tween annual mean values with temporally variable and constant emissions can be up to a few ppm in the hot spot regions, but are mostly below 1 ppm elsewhere. This implies that temporal variability of fossil fuel emissions needs to be accurately represented for realistic simulations, confirming the results of Zhang et al. (2016). It is also important for reliably detecting fossil fuel emission changes from specific sources since different sources have different temporal profiles.

Simulating fossil fuel emissions from different countries and sectors suggests that the major part of the signal near the surface remains in the country of origin. Ground-based in situ observations are thus most sensitive to fossil fuel emissions from the country where they are located. A different picture emerges for column averaged dry air mole fractions ($X\mathrm{CO_2}$) as measured by satellites, for which the signal is much more dispersed. Only over Germany, the contribution from emissions within the country is larger than 50%, whereas over France the signal from neighboring countries dominates (66%). An important reason for these contrasting results seems to be the differences in electricity production, which mostly relies on nuclear power in France but on fossil fuels in its neighboring countries including Germany, UK and Italy. Over small countries such as Switzerland or the Netherlands, the contribution from abroad is typically the dominating component. Among all the processes, fossil fuel emissions from power plants contributes the most (approx. one third) to the total fossil fuel signal of $\mathrm{CO_2}$ both at the surface and in the column. However, the power plant signal at the surface is likely overestimated in our simulations, since all emissions were released into the lowest model level without considering the true elevation of the source. The signal from power plant emissions has a pronounced and distinct spatial pattern that provides us an opportunity to discern changes in from power plant emissions from changes in other sources.

Based on a number of sensitivity studies, we show that reductions in fossil fuel emissions not only leave a distinct signal in the time mean distribution of atmospheric $\mathrm{CO_2}$, but also in its temporal variability. This opens potentially additional ways to detect and verify emission reductions. But this opportunity exists primarily for surface based measurement networks, while the satellite based systems that measure the column-averaged $X\mathrm{CO_2}$ will see too small changes, in general, relative to their current measurement capabilities. An important exception are a few hotspot sites, where the satellites will be able to detect fairly modest changes of about 30% when assuming an accuracy of the satellite observations of 0.5 ppm.

As both satellite and surface measurements have advantages and disadvantages, combining surface measurements with satellite data and increasing the frequency and coverage of the latter will be the optimal path forward to enhance the possibility of detecting future changes in fossil fuel emissions.

*Acknowledgements.* This study was funded by the Swiss National Foundation (SNF) as part of the CarboCount CH Sinergia Project (Grant Number: CRSII2 136273). Additional funding was provided by ETH Zürich (YL and NG) and Empa (DB). We thank the Center for Climate Systems Modeling (C2SM) and especially Anne



Roches for providing extensive support. We acknowledge MeteoSwiss for the provision of their operational COSMO analysis products. The computations were done on the supercomputers of the Swiss National Super-

770 computing Centre (CSCS). We are very thankful to the principal investigators responsible for the Mace Head (MHD) station in Ireland, for the SNO-ICOS-France station Puy de Dome (PUY) in France, for the Hegyhatsal station (HUN) in Hungary, and for the Cabauw station (CBW) in the Netherlands for making their data publically available. We are also grateful to Dr. Sander van Der Laan and Dr. Felix Vogel for sharing their estimates of the fossil fuel component at Lutjewad and Heidelberg, respectively. Furthermore, we are very grateful to

775 Prof. Woulter Peters, Wageningen University, for providing the $CO_2$ global model boundary conditions and to Dr. Christoph Gerbig for providing the VPRM biospheric flux data.





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



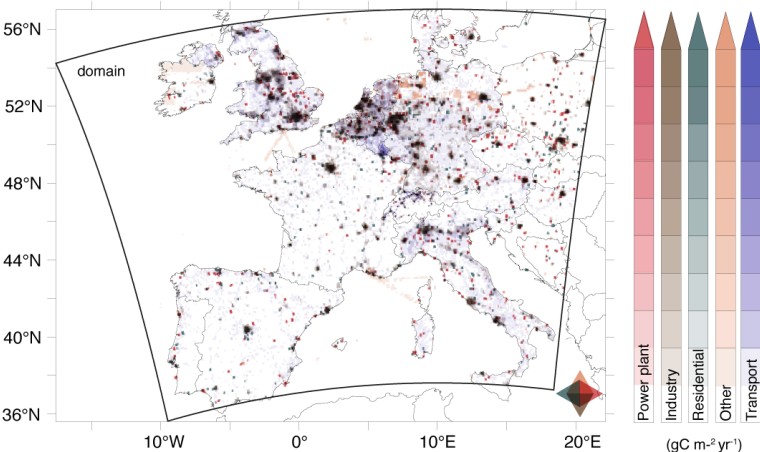

Figure 1: Map of the fossil fuel emissions used in this study. Also depicted is the domain of the COSMO-7 setup employed here. Shown in transparent color are the fossil fuel $CO_2$ emissions for different sectors in units of $gC\,m^{-2}\,yr^{-1}$. The colors from the different sector blend to a darker color when they are co- located as shown by the color mixing star at the bottom right.

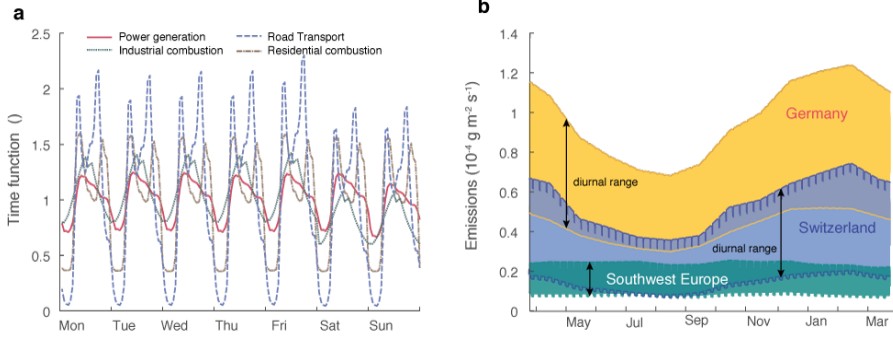

Figure 2: Time dependence of fossil fuel $CO_2$ emissions for different sectors and countries. (a) Time functions for the diurnal and weekly emissions for four sectors. (b) Annual evolution of the $CO_2$ emission intensity for three different countries or group of countries. Shown are the daily minima and maxima for each country.





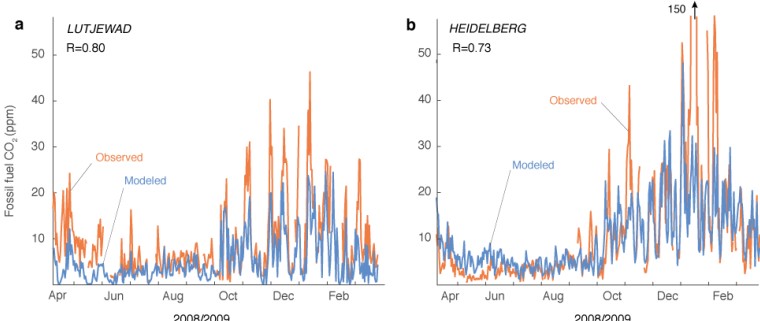

Figure 3: Comparison between modeled and observation-based estimates of the fossil fuel $CO_2$ component. (a) Comparison at the Lutjewad site in the Netherlands (LUT, 6° 21'E, 53° 24'N, 1 m a.s.l.) (Van Der Laan et al., 2010; Bozhinova et al., 2014). (b) Comparison at Heidelberg (HEI, 49.417° N, 8.675° E, 116m a.s.l.) (Levin and Karstens, 2007). The observational estimates are based on concurrent observations of CO and $^{14}CO_2$.

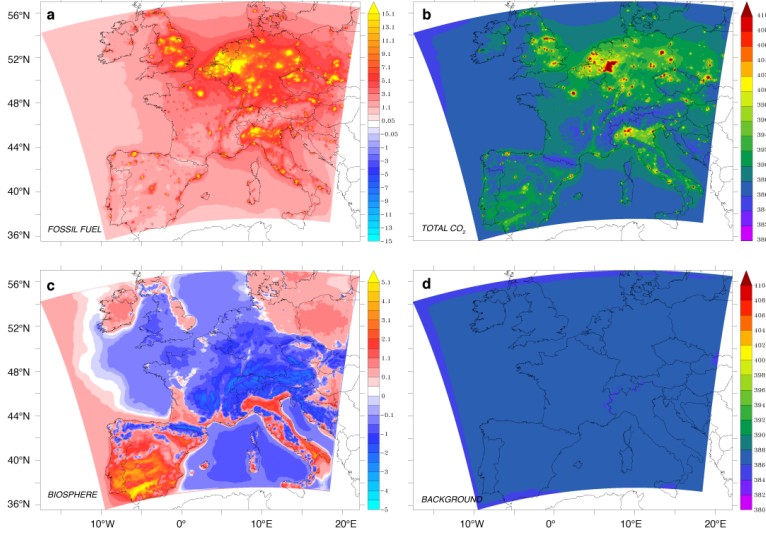

Figure 4: Maps of the model simulated annual mean components of atmospheric $CO_2$ in the surface layer (10 m above ground). (a) fossil fuel component, (b) total atmospheric $CO_2$, (c) terrestrial biosphere component, and (d) background $CO_2$ component. The results are shown as dry air mole fraction with units of ppm.





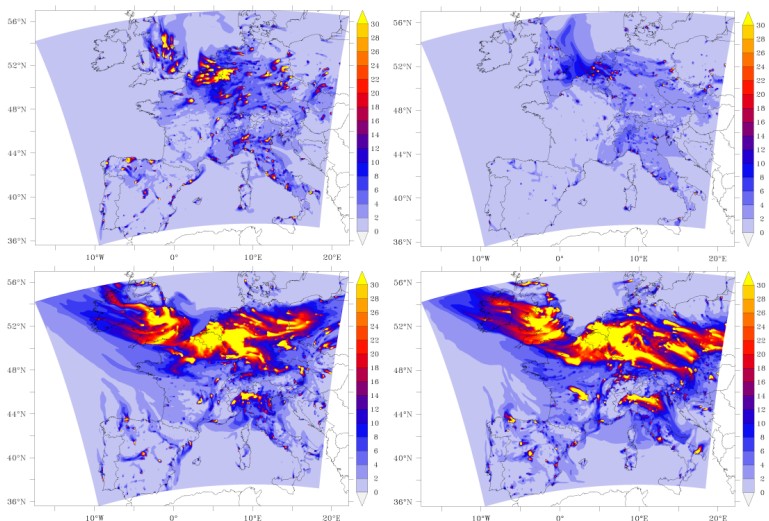

Figure 5: Instantaneous snapshot of the model simulated fossil fuel $CO_2$ in the surface layer. (a) Snap shot on July 1st at 06 00 GMT, (b) as (a) but at 18 00 GMT, (c) snapshot on January 1st at 06 00 GMT, (d) as (c) but at 18 00 GMT.

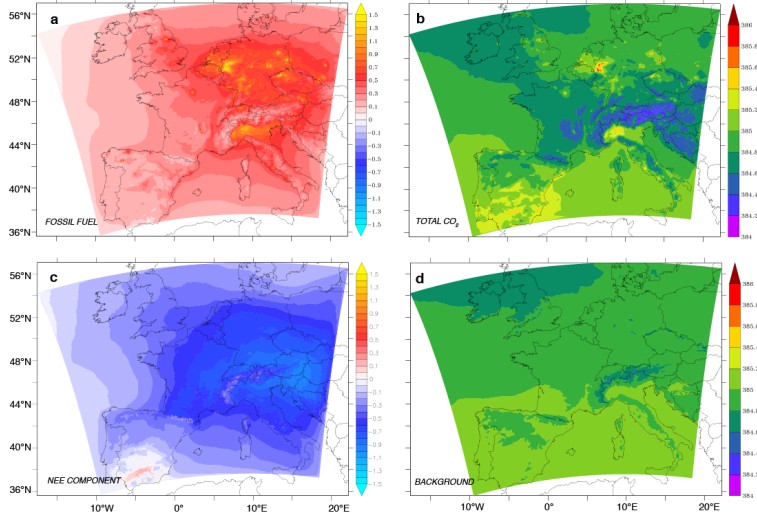

Figure 6: As Figure 4, but for whole air column averaged dry air mole fraction in units of ppm .





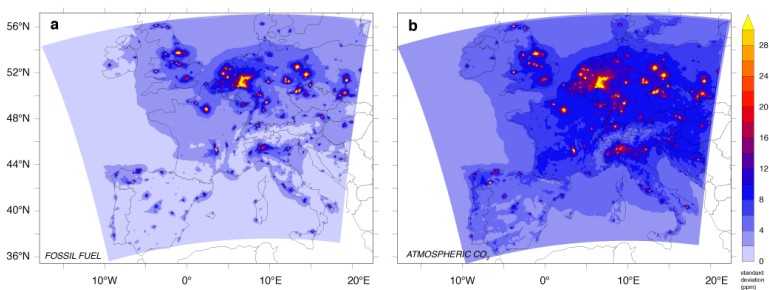

Figure 7: Maps of the annual standard deviation of (a) the fossil fuel component and (b) atmospheric $CO_2$ in the surface layer.

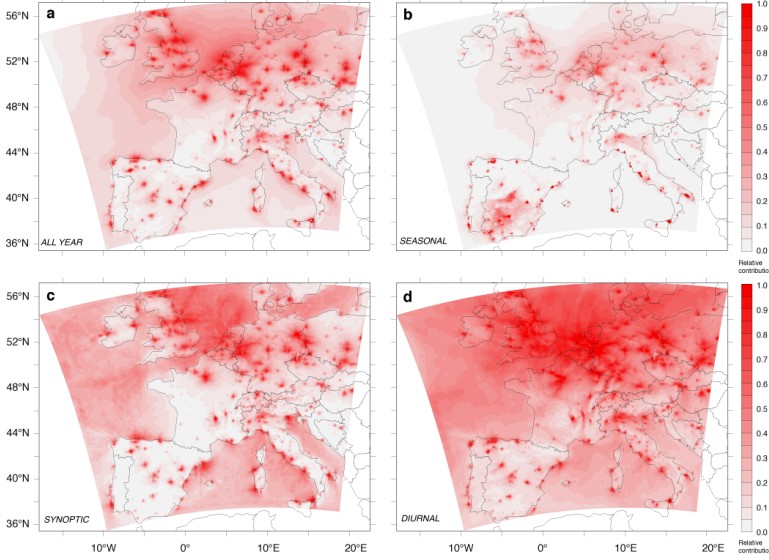

Figure 8: Maps of the contribution of fossil fuel $CO_2$ variability to total atmospheric $CO_2$ variability on various timescales in percent. (a) Contribution over all timescales; (b) contribution for the seasonal timescale only; (c) contribution for the synoptic timescale only; (d) contribution for the diurnal timescale only.



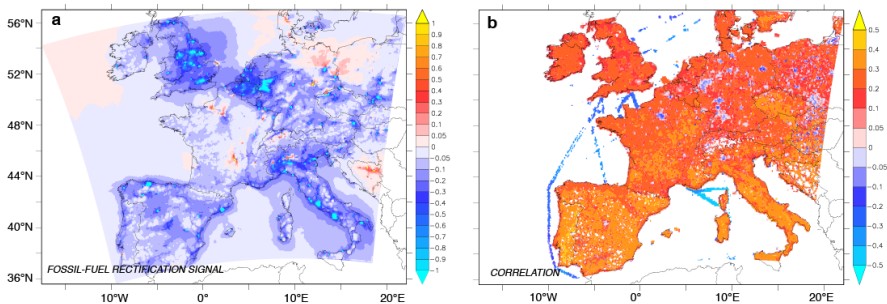

Figure 9: Maps of the impact of the consideration of time-varying fossil fuel emissions. (a) Difference in annual mean surface $CO_2$ between the case with time varying and time-constant fossil fuel emissions. This difference represents the fossil fuel rectification effect. (b) Linear correlation between the fossil fuel emissions and the height of the planetary boundary layer height in the COSMO-7 model. Pixels with emissions smaller than 0.06 gC m$^{-2}$ yr$^{-1}$ are not plotted. The positive correlation implies high emissions when the PBL is deep, and vice versa. Most of this correlation stems from the diurnal time-scale, but the correlation is enhanced through the (mostly) positive correlation also on seasonal timescales.

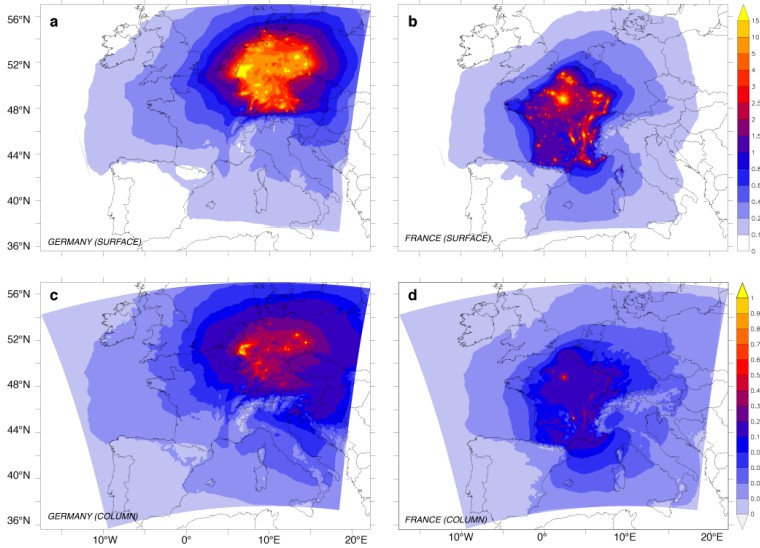

Figure 10: Maps of the annual mean fossil fuel $CO_2$ generated by different countries/regions. (a) Surface pattern created by the emissions from Germany, (b) as (a), but for the France. (c) Column averaged pattern created by the emissions from Germany, and (d) as (c), but for France.

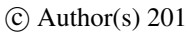



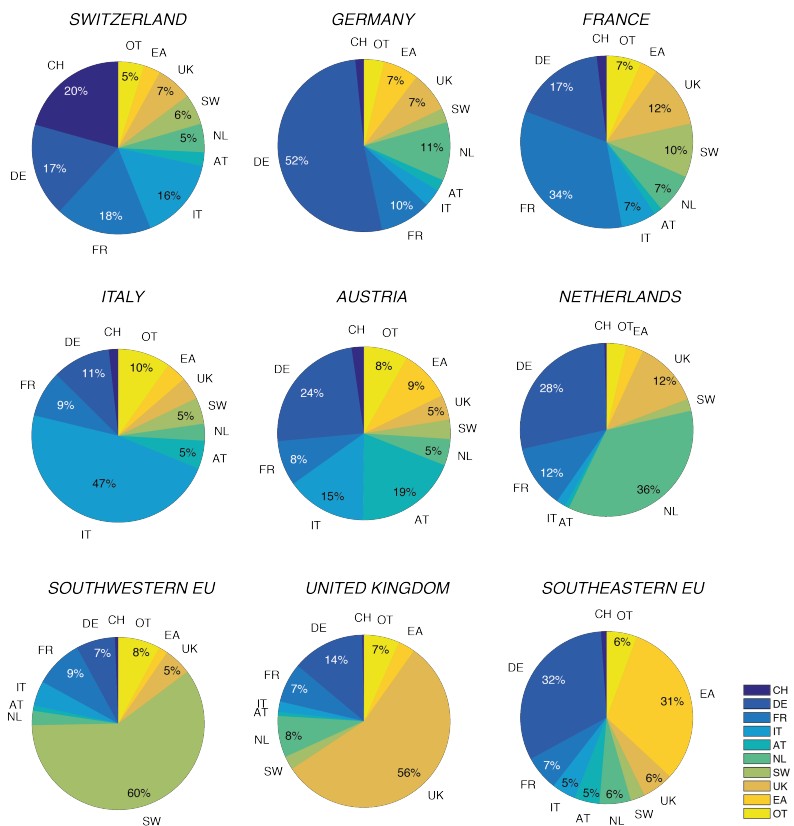

Figure 11: Pie charts depicting the origin of the fossil fuel $CO_2$ signal for each country/region. The percentages represent the contribution of each country/region of origin to the total fossil fuel signal in the averaged over the air column. The pie chart for Switzerland reveals, for example, that only 20% of the fossil fuel $CO_2$ signal over its territory stems from its territorial emission. Here, CH: Switzerland; DE: Germany; FR: France; IT: Italy; AT: Austria; NL: Netherlands; SW: countries in southwest of the domain; UK: United Kingdom; EA: countries in eastern domain; OT: the rest of countries.



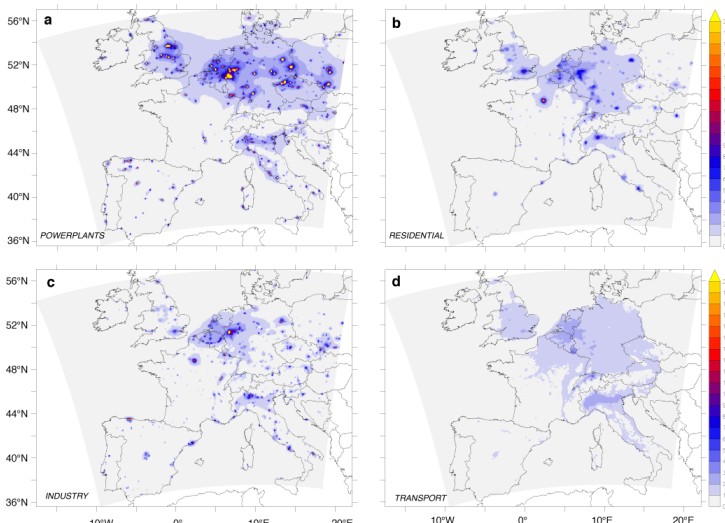

Figure 12: Maps of the annual mean surface fossil fuel $CO_2$ stemming from different sectors in units of ppm. (a) fossil-fuel fired power plants, (b) residential heating, (c) industrial processes, and (d) road transportation.

Table 1: Evaluation of COSMO-7 based simulations of the atmospheric $CO_2$ concentration at 4 European sites (locations are shown in Figure 1). The comparison are shown for the 3 hourly means between 12 to 18 PM local time for the period April 2008 through April 2009. m.s.a.g.is the height above ground or relative height.

| Station | characteristics | m.s.a.g(m) | S.T.D. obs(ppm) | S.T.D. mod(ppm) | Correlation | Bias(ppm) |
|---|---|---|---|---|---|---|
| Cabauw (CBW, Netherlands) | tower | 20 | 11.46 | 11.80 | 0.78 | 0.72 |
| Cabauw (CBW, Netherlands) | tower | 60 | 10.86 | 11.06 | 0.77 | 0.27 |
| Cabauw (CBW, Netherlands) | tower | 200 | 9.35 | 8.19 | 0.74 | 0.88 |
| Puy de Dome (PUY, France) | mountain top | 10 | 7.83 | 7.65 | 0.75 | 0.85 |
| Hegyhatsal (HUN, Hungary) | continental | 10 | 12.08 | 9.42 | 0.8 | 4.0 |
| Hegyhatsal (HUN, Hungary) | continental | 48 | 11.51 | 9.32 | 0.8 | 4.04 |
| Hegyhatsal (HUN, Hungary) | continental | 115 | 10.69 | 8.72 | 0.8 | 3.86 |
| Mace Head, (MHD, Ireland) | coastal | 15 | 6.26 | 3.87 | 0.81 | -0.34 |





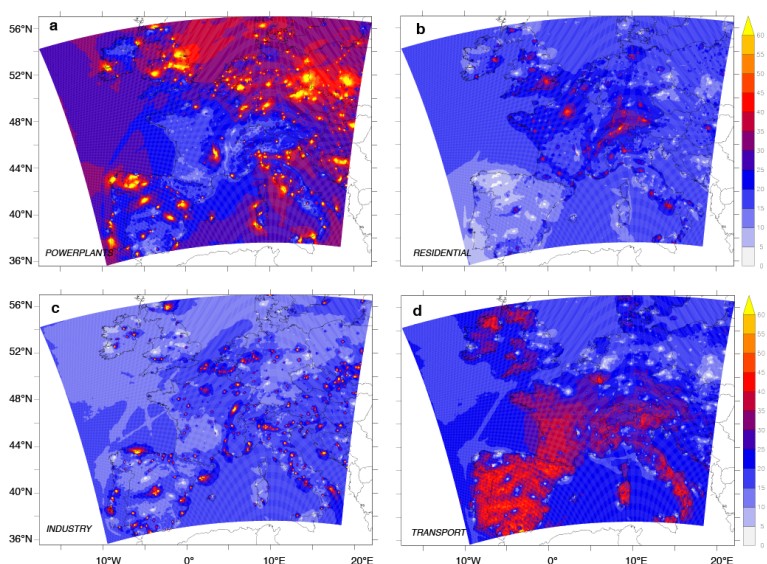

Figure 13: Maps of the annual mean relative contribution of each sector to the total surface fossil fuel $CO_2$. a) fossil-fuel fired power plants, (b) residential heating, (c) industrial processes, and (d) road transportation.





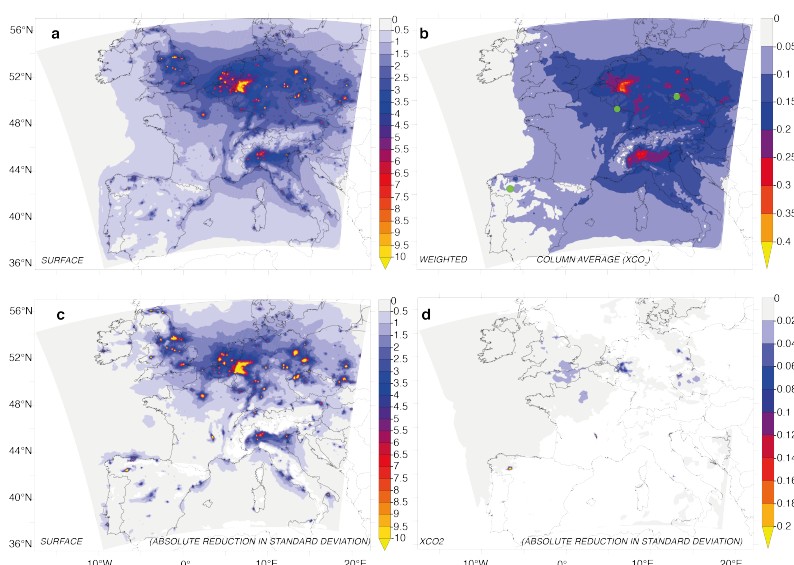

Figure 14: Changes in annual mean atmospheric $CO_2$ and its standard deviations resulting from a 30% reduction in the fossil fuel emissions from all sectors. (a) Change in surface mean $CO_2$. (b) Change in the column averaged $CO_2$, i.e., $XCO_2$. (c) Change in the standard deviation of surface $CO_2$ (all seasons). (d) Change in the standard deviation of the column averaged $CO_2$, i.e., $XCO_2$. The standard deviation refers to the differences of the afternoon data (at 1:00 PM) to the annual afternoon average.





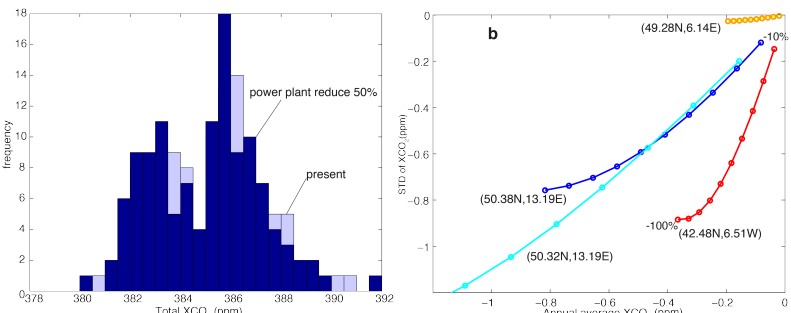

Figure 15: Impact of reductions in power plant emissions on the mean and standard deviation of the fossil fuel $CO_2$ signal . (a) probability density distribution of the surface atmospheric $CO_2$ for the present and for a case when the power plant emissions were reduced by 50% at a site in eastern Germany (50.32°N,13.19°E). (b) Relationship between the changes in the mean and the standard deviation of the column averaged $CO_2$ for a given reduction in power plant emissions, with different color representing representing different sites with different characteristics in their response to this reduction in emission: Blue (50.32°N, 13.19°E), Cyan (50.32°N, 6.59°E), Red (42.48°N, 6.51°W), Orange (49,28°N, 6.14°E) (Locations shown in Figure 14b with green circles).