# Peer review of "Spatiotemporal patterns of the fossil-fuel CO2 signal in central Europe: Results from a high-resolution atmospheric transport model"

_Atmospheric Chemistry and Physics, 2017_

## Referee Comment (RC1) · Anonymous Referee #1 · 8 May 2017

**General:**
The paper rises very important and actual topic how the fossil-fuel related contributions to the global CO2 distribution can be separated from the other CO2-related sources and sinks. Using a validated model, the spatiotemporal patterns of CO2 are discussed which might help to discern the anthropogenic contributions from the other possible sources of variability whereas both, in situ and satellite-based strategies are considered. The paper is well-written and gives new and important insights. However, there is still one (major) point which have to be discussed before this paper can (and should) be published.

[Figure]

**Major points:**

1. My main concern is related to the description/handling of the background contribution (L212-217) as derived from the CarbonTracker data set and used to prescribe the boundaries. Do you use mixing ratios or fluxes at these boundaries? Because CarbonTracker data contains $CO_2$ mixing ratios for sources within and without your model domain (which is central and southern Europe) I have a feeling that your boundary conditions contain both information although the $CO_2$ contribution from the inside should be resolved by your model. I am not sure but it looks like for the boundary conditions you would take into account the "same type of information" twice: from your simulation and from the CarbonTracker data. At least, would be good to have a more detailed description, how you handle boundary conditions and especially this point.

**Minor points:**

1. L14
   "...their co-variance leads to a fossil-fuel diurnal rectifier.." - For "no-experts" difficult to understand.

2. L82
   Maybe you should explain with 1-2 sentence what is "rectification".

3. L113
   You mean "potentially reduced emissions".

4. L155-57
   It would be nice to understand this formula without checking other literature. What is $K$? I think $K$ should be the highest level of the model ?. Would be good to have this formula as a separate equation.

5. L349
   Maybe: "by the diurnal and seasonal variations" and remove the last part of the sentence.

6. L385
   ...uniform negative distribution for $XCO_2$ in Fig. 6c contrasts...

7. L445
   ...in particular, what is the contribution of diurnal (and seasonal)...

8. L490
   ...Figure Fig....

9. L724
   "...up to 110%...". Not clear what does it exactly mean. Please explain.

10. Figure 4
    Figure 4a shows the anomaly and not the absolute value. You should explain how this anomaly is defined (in caption and main text). Same for Figure 4b. Figure 4d does not show any structure (maybe you should change the color bar). It is not clear for me what should I see. Impact of the boundaries on the main domain of your model? See also my major point.

11. Figure 5
    Please add notation: a), b), c) d).

12. Figure 6
    Please explain/define the anomalies

13. Figure 8
    Please add that all panels are for the surface layer (10m)

14. Figure 9
    There are some lines which look like ship connections. Maybe you would like to explain it.

15. Figure 11
    This figure is not mentioned in the text. Also you write sometimes "Figure" and sometimes "Fig."

16. Figure 13a
    There are enhanced values in north-east (over the North Sea). Maybe you should explain this feature in the text.

---

## Referee Comment (RC2) · Anonymous Referee #2 · 29 May 2017

**Review of Liu et al.: Spatiotemporal patterns of the fossil-fuel CO2 signal in central Europe: Results from a high-resolution atmospheric transport model**

General comments: the authors firstly presented the spatiotemporal patterns of the atmospheric fossil fuel CO2 concentration over the central and southern Europe, and then conducted a sensitivity study on the impact of fossil fuel emission on the atmospheric CO2 concentration varying with emission reduction and regions.

Overall, this study is interesting, fits the scope of ACP, and shows enough findings to be published. However, it is quite confusing to me sometimes that what the authors referring is fossil fuel emissions or fossil fuel CO2 signals (concentration). One is a flux based concept, another is mixing ratio. The authors need to make it clear.

The authors conducted a reasonable amount of the statistics to support their findings, i.e., Table 1. However, based on their results, they used quite a few statements like "good agreement", "agree well", etc.  These kinds of statements have no reference in the entire manuscript: no inter-comparison with other studies, and/or no significant test, which makes these statements are too vague and scientifically meaningless.

It will be very helpful if the authors list more information in the figure caption. For example, it's not clear to me that which figure is based on the all time series, which based on the daytime afternoon. The bottom line is that readers can understand the figures without checking the text.

Overall, the authors did model evaluation (comparison to observations) based on the daytime values but did the annual mean based on the all time series. The models are struggled with simulating the nighttime CO2 in general. The all time series will very likely bias the entire results of annual mean.

There are quite a lot details needed to be fixed and listed below as the specific comments. I would recommend the authors to work on the consistency of the statement and correct those comments before they resubmit the manuscript.

Specific comments:
1. Line 22, what do you mean the last sentence of the abstract?
2. Line 50, Newmant et al., (2016, ACP) also studied the fossil fuel section based on the observation approach.
3. Line 63, it can't be true that the model dynamic is fully resolved. The subgrid processes should be parameterized.
4. Line 87-93, the authors is explaining the possibility to detect the fossil fuel signal, which I was assuming that it referred CO2 concentration. However, the two approaches – bottom-up and top-down – are used to quantify CO2 fluxes in general. This confusion appeared a lot throughout the entire paper and needs to be fixed.

5. Line 132, in the model setup, the flux components are called surface flux and, can be considered as surface boundary condition. The prescribed global model output providing the advection on the boundaries of model domain is lateral boundary conditions. The statement regarding these drivers is not clear to me.
6. Line 170, the authors used time function to scale the monthly emission in time. I recommend the authors to make a figure that shows the time series of the scaled emission components and the total emission on top of Figure 2a. I also would like to see how the authors reconcile the discontinuity between weeks and between months.
7. Line 207, a table to describe all of the fossil fuel tracers will be very helpful for readers to follow.
8. Line 207 and 226 contradict each other. Please double check how many fossil fuel tracers used, and correct it if it's not.
9. Line 225, see item 5 above.
10. Line 235, does R>0.7 mean significant correlation?
11. Line 248, "underestimate" or overestimate? All of the bias values in Table 1 are positive but the one at Mace Head.
12. Line 248 – 259, I can't find those value in Tables. It's confusing.
13. Line 261, in Table 1 the STD for the model are ~4 ppm and ~9ppm at Mace Head and Hungary, respectively.  Please correct them and carefully check the values are stated in the manuscript.
14. Line 300,  see item 10.
15. Line 314, what is the criteria for "the good to excellent evaluation"? Could the authors quantify them?
16. Line 317 – 319, "It is particularly ….". " The presence of an overall …" These two sentence contradict each other.
17. Line 320, the statistics were made sometimes based on the daytime values sometimes based on the all time series. Overall, the authors did model evaluation (comparison to observations) based on the daytime values but did the annual mean based on the all time series. The models are struggled with simulating the nighttime $CO_2$ in general. The all time series will very likely bias the entire results of annual mean.
18. Line 333, more explanation needed on "suggesting a somewhat limited efficiency of atmospheric transport and mixing to disperse the signal laterally". More explanation needed.
19. Line 334, a map of terrain for the region of interest will be very helpful.
20. Line 360, "while the biospheric signal is stronger".  Biospheric signal can be positive and negative, which will cause completely opposite effect on the total CO2. Please clarify it.
21. Line 369, the column averaged values are smaller than the surface ones, mainly because the signals are at surface and they are diluted in the column as averaged out. The statement is incorrect.
22. Line 400, what does lateral gradient mean specifically here?
23. Line 419-428, Figure 8 and the relevant statement doesn't make sense to me. I don't understand that why the contribution of each component (b-d) would be larger than the total (a).

24. Line 487-555, the authors made misleading statement regarding the relationship between meteorological matric (such as PBL, etc.) and fossil fuel emission. Fossil fuel emission is strongly correlated to the anthropogenic activities. The increase and decrease of the fossil fuel emission is not affected by transport and/or mixing; the atmospheric co2 concentration is. At the beginning, I thought it ("emission") is a typo, which is an easy fix. However, I realized that this relationship is not clear at all to the authors when I saw Figure 9b.

25. Line 670, the authors selected three location to do further analysis for Figure 15. Why these three? Please explain.

26. Line 700, yes, the authors were being optimistic as they admitted. All of the analyses were based on the model results. The cloud contamination was considered in the discussion. However, the more important errors are very likely caused by model transport errors and error estimate of biogenic flux. It will be great if the author can expend the discussion on the impact of those error on their findings.

27. Line 724, where does "110%" come from? Is it a new? The authors shouldn't include any new results in the conclusions.

28. Figure 1, please put the labels on the color bars.

29. Figure 2, I would like to see the time series of the scaled emission components and the total emission.

30. Figure 4, 1) enlarge the font size. 2) what year is it? 3) night time model results included? If so, 10 m above ground is too late. Models are struggled at nighttime, especially for such low level. 4) Keep the color bars in the scale and same range for a and c, b and d.

31. Figure 5, the figure labels (a, b, c, d) are missing.

32. Figure 6, use the same color scale as Figure 4 does.

33. Figure 8 doesn't make sense to me as I pointed out before. The contribution of each component can't larger than the total.

34. Figure 9a, which minus which? Figure 9b doesn't make sense to me. The fossil fuel emission is related to anthropogenic activities; it won't be affected by transport and/or mixing. If it is concentration instead of emission. The correlation of the concentration and PBL height should be negative. It's confusing.

35. Figure 11, what countries are they exactly included in SW?

36. Table 1, it's a confusing statement that "the 3 hourly means between 12 – 18 PM"

37. Figure 13, enlarge the font size. Does it in percentage? Can the authors overlay the wind field on top?

38. Figure 14, it's not clear to me to look at the standard deviation changes between two cases.

39. Figure 15, I can only find three green dots in Figure 14b, but Figure 15b shows four lines. Please make correction accordingly.

---

## Author Comment (AC1) · 14 Jul 2017

**Spatiotemporal patterns of the fossil-fuel $CO_2$ signal in central Europe: Results from a high-resolution atmospheric transport model**

Yu Liu, Nicolas Gruber, and Dominik Brunner

We would like to thank the reviewer for the efforts to review this manuscript and the thoughtful comments and remarks. All of the referee's comments have been carefully examined and addressed in the revised manuscript.
* * *
**Referee #1**

**General comments:**

*My main concern is related to the description/handling of the background contribution (L212-217) as derived from the CarbonTracker data set and used to prescribe the boundaries. Do you use mixing ratios or fluxes at these boundaries? Because CarbonTracker data contains CO2 mixing ratios for sources within and without your model domain (which is central and southern Europe) I have a feel- ing that your boundary conditions contain both information although the CO2 contribution from the inside should be resolved by your model. I am not sure but it looks like for the boundary conditions you would take into account the "same type of information" twice: from your simulation and from the CarbonTracker data. At least, would be good to have a more detailed description, how you handle bound- ary conditions and especially this point.*

**Authors:** We thank the reviewer for this comment and the associated questions, as our description of our handling of the boundary conditions was indeed not detailed enough. This will be rectified in the revised version.

Concretely, we are using a relaxation boundary condition for all tracers, although with slightly different details depending on whether the tracer enters the domain from the outside (e.g., the background $CO_2$) or whether it has sources and sinks inside

the domain (e.g., the different fossil fuel tracers and the terrestrial biosphere $CO_2$). In the former case, i.e., for the background $CO_2$, we are using a "full" relaxation boundary condition. This means that we are restoring the modeled mixing ratio toward the value provided by CarbonTracker across a transition zone, with the relaxation increasing in strength from the inner to the outer border of this zone. In COSMO, this option is provided by the "T_RELAX_FULL" switch. In the latter case, i.e., for fossil fuel and the biosphere $CO_2$, we are using a partial relaxation. In this case, the tracer is relaxed to the boundary condition only at the outermost grid cells of the domain and only when the wind is directed toward the inside of the domain (in COSMO, this option is provided by the switch "T_RELAX_INFLOW"). Through this option, we avoid creating a situation where the zero concentration boundary condition is propagated (erroneously) against the flow back into our domain. While we consider this partial relaxation to be the better option for our fossil fuel and biosphere tracers, we suspect at the same time that the impact of this choice is relatively small, i.e., that the results would not differ much had we selected the full relaxation option, because the domain is large in comparison to the transition zone.

Our choice of boundary condition coupled with our separating the total $CO_2$ into its different components has indeed some implications for the "double counting" that the reviewer correctly identifies as an issue to pay attention to. In an ideal case, we would have run the same set of tracers in both the global CarbonTracker and the regional COSMO simulations with two way coupling, as this would not have caused any inconsistencies at all. In our one-way coupling mode plus CarbonTracker incorporating also the fossil fuel signal from Europe into its background, there is indeed a chance for a mismatch between what is considered "background" and what is considered a "fossil fuel signal". This becomes evident when considering an air parcel containing a fossil fuel signal from within Europe to leave the domain and then to return back into our domain. In such a case the fossil fuel signal loses its identity by leaving the domain, and becomes a background signal in CarbonTracker's modeling the region outside of Europe. This signal would then enter the domain again as a background signal through our restoring the background $CO_2$ toward CarbonTracker's results. Thus, our handling of the boundary conditions does not lead to a double counting, but a potential underestimation of the total fossil fuel signal and a potential overestimation of the background CO2.

In response, we added the following texts to the methods

"At the lateral boundaries, we employ a partial relaxation boundary condition for these 17 tracers. In such a partial relaxation, the tracer is relaxed to the boundary concentration only at the outermost grid cells of the domain and only when the wind

is directed toward the inside of the domain (in COSMO, this option is provided by the switch "T_RELAX_INFLOW"). Since we are interested in the fossil fuel signal emanating from emissions in Europe only, the lateral boundary concentration was set to zero. Through this option, we avoid creating a situation where the zero concentration boundary condition is propagated (erroneously) against the flow back into our domain."

"[...]For this tracer, we use a "full" relaxation boundary condition. This means that we are restoring the modeled mixing ratio toward the value provided by CarbonTracker across a transition zone consisting of around 13 grid cells, with the relaxation increasing in strength from the inner to the outer border of this zone. In COSMO, this option is provided by the "T_RELAX_FULL" switch."

"[...] The lateral boundary conditions for these two tracers were handled the same way as those for the fossil fuel signal, i.e., a partial relaxation toward a zero concentration at the boundary."

**Specific comments:**

*1.L14 "...their co-variance leads to a fossil-fuel diurnal rectifier.." –*

*For "no-experts" difficult to understand.*

**Authors:** We changed the sentence to "The covariance of the fossil fuel emissions and atmospheric transport on diurnal timescales leads to a diurnal fossil-fuel rectifier effect as large as 9 ppm compared to a case with time-constant emissions.."

*2.L82 Maybe you should explain with 1-2 sentence what is "rectification".*

**Authors:** To make it clearer, we extended our explanation to " Of particular relevance are the diurnal and the seasonal changes in emissions, since they tend to co-vary with atmospheric transport, which can lead to annual mean atmospheric $CO_2$ concentration gradients that differ from those attained if the emissions were held constant. This difference, which arises solely from the co-variation between fluxes and transport, is called a "rectification effect" in analogy to the rectification of an AC voltage in an electrical circuit by a diode ".

*3.L113  You mean "potentially reduced emissions".*

**Authors:** Yes. Changed to "potentially reduced emissions"

*4.L155-57 It would be nice to understand this formula without checking other literature. What is K? I think K should be the highest level of the model ?. Would be good to have this formula as a separate equation.*

**Authors:** K is the total number of vertical model levels (K=60). We added this explanation and also formatted the equation now as a separate equation in the text.

*5.  L349  Maybe: "by the diurnal and seasonal variations" and remove the last part of the sentence.*

**Authors:** The objective of the sentence is to demonstrate the importance of the diurnal transport. Thus, we prefer to make no change to the sentence.

*6.L385  ...uniform negative distribution for $XCO_2$ in Fig. 6c contrasts...*

**Authors:** Done.

*7.L445  ...in particular, what is the contribution of diurnal (and seasonal)...*

**Authors:** Done.

*8.L490  ...Figure Fig....*

**Authors:** Done

*9.L724  "...up to 110%...". Not clear what does it exactly mean. Please explain.*

**Authors:** We rephrased this sentence to:

"In some places, it even contributes significantly to the total (including background) CO2, particularly in large urban centers and along power plant plumes."

*10.    Figure 4 Figure 4a shows the anomaly and not the absolute value. You should explain how this anomaly is defined (in caption and main text). Same for Figure 4b. Figure 4d does not show any structure (maybe you should change the color bar). It is not clear for me what should I see. Impact of the boundaries on the main*

*domain of your model? See also my major point.*

**Authors:** This must be a misunderstanding, since Figures 4a (fossil fuel component) and c (biosphere) are showing the actual (absolute) value of these components and not their anomalies. The confusion might emerge from the way the lateral boundary conditions are set for these two tracers, i.e., they are set to zero, as their variations reflect just the sources minus sinks within the domain. We hope that our response and changes to the text in response to the major comment by this reviewer takes also care of this comment.

Figure 4d shows the "background $CO_2$" component, i.e., that part of the variation in the total $CO_2$ that is determined through the lateral boundary conditions. We use the same color scale for both panels b and d to show that the variations imprinted by the background has a small impact on the spatial pattern of the total $CO_2$ concentration within the domain.

11.  *Figure 5  Please add notation: a), b), c) d).*

**Authors:** Done.

12.  *Figure 6  Please explain/define the anomalies*

**Authors:** The same as Figure 4. These panels show the actual (absolute) values, not the anomalies. Again, we hope that our additional text in response to the main comment by this reviewer clarifies this issue.

13.  *Figure 8  Please add that all panels are for the surface layer (10m)*

**Authors:** We modified the figure caption to read " Maps of the contribution of fossil fuel CO2 variability to total atmospheric CO2 variability within the lowest model layer (0-20 m, center at 10 m) on various timescales in percent."

14.  *Figure 9 There are some lines which look like ship connections. Maybe you would like to explain it.*

**Authors:** These lines are indeed due to the fossil fuel emissions stemming from marine transportation. A comment was added to the caption to this effect:

"The negative correlations over the ocean stem from the fossil fuel emissions by ships."

15. *Figure 11 This figure is not mentioned in the text. Also you write sometimes "Figure" and sometimes "Fig."*

   **Authors:** We changed Fig. into Figure throughout the manuscript. Figure 11 is actually discussed in the text, but we were erroneously referring to Figure 10 instead of Figure 11. This was fixed.

16. *Figure 13a There are enhanced values in north-east (over the North Sea). Maybe you should explain this feature in the text.*

   **Authors:** We added the following text:

   "Owing to the dominance of this mode of electricity production in northern Europe, this signal is particularly strong there. This is most evident over the North Sea, where the advection of the emitted $CO_2$ from the power plants in the UK and the Netherlands creates a particularly visible plume over the ocean."

---

## Author Comment (AC2) · 17 Jul 2017

**Spatiotemporal patterns of the fossil-fuel CO$_2$ signal in central Europe: Results from a high-resolution atmospheric transport model**

Yu Liu, Nicolas Gruber, and Dominik Brunner

We would like to thank the reviewer for the efforts to review this manuscript and the thoughtful comments and remarks. All of the referee's comments have been carefully examined and addressed in the revised manuscript.

**Referee #2**

**General comments**:
*The authors firstly presented the spatiotemporal patterns of the atmospheric fossil fuel CO2 concentration over the central and southern Europe, and then conducted a sensitivity study on the impact of fossil fuel emission on the atmospheric CO2concentration varying with emission reduction and regions.*
*Overall, this study is interesting, fits the scope of ACP, and shows enough findings to be published.*

> **Authors:** Many thanks for the positive remarks.

*However, it is quite confusing to me sometimes that what the authors referring is fossil fuel emissions or fossil fuel CO2 signals (concentration). One is a flux based concept, another is mixing ratio. The authors need to make it clear.*

> **Authors:** The reviewer is absolutely correct in that we clearly need to distinguish between the two. We went through the text again and ensured that this is taken care of. Specifically, we added "signal" or "concentration" whenever we refer to the atmospheric concentration and "emission" whenever we refer to the fluxes.

*The authors conducted a reasonable amount of the statistics to support their findings, i.e., Table 1. However, based on their results, they used quite a few statements like "good agreement", "agree well", etc. These kinds of statements have no reference in*

*the entire manuscript: no inter-comparison with other studies, and/or no significant test, which makes these statements are too vague and scientifically meaningless.*

**Authors:** We agree with the reviewer that care needs to be taken when using such qualitative statements. But we feel that we had (mostly) adhered to common scientific standards, e.g., those established by IPCC, by using "good/well" when the probability exceeds 66% (likely in the parlance of IPCC) and very good/very well if it exceeds 90% (very likely for IPCC). We have checked the paper again and made sure that we use such statement only when they are fully justified.

*It will be very helpful if the authors list more information in the figure caption. For example, it's not clear to me that which figure is based on the all time series, which based on the daytime afternoon. The bottom line is that readers can understand the figures without checking the text.*

**Authors:** We added the time period to the caption of almost all figures. We extended the caption also with regard to other elements, such as the layer considered in the plots (also responding to a comment by the first reviewer).

*Overall, the authors did model evaluation (comparison to observations) based on the daytime values but did the annual mean based on the all time series. The models are struggled with simulating the nighttime CO2 in general. The all time series will very likely bias the entire results of annual mean. There are quite a lot details needed to be fixed and listed below as the specific comments. I would recommend the authors to work on the consistency of the statement and correct those comments before they resubmit the manuscript.*

**Authors:** The reviewer is correct that current generation atmospheric models tend to deviate more strongly from observations at night compared to the day, largely owing to the difficulties of correctly modeling the night time shoaling of the atmospheric boundary layer. This is why we were evaluating the model simulated $CO_2$ against daytime observations only in the paper. But while we feel that this does not lead to any inconsistencies with our showing later results reflecting a true daily average, we nevertheless decided to make all evaluations and results consistent, i.e., being based on true daily averages. A key reason for this choice is also that the difference is actually not particularly large.

We thus delete the sentence (line 236/37)

"In order to minimize the impact of local influences, we use the average CO2

concentrations between 12:00 and 18:00 local time, i.e., the time of day of maximum vertical mixing."

and add "daily averaged" in front of "measurements" to read
"by comparing them to daily averaged measurements" (line 231)

Furthermore, we changed all values in Table 1. No change is needed for Figure 3, as this was already based on a daily average.

*Specific comments:*
1. *Line 22, what do you mean the last sentence of the abstract?*

   **Authors:** It means that changes in the standard deviation of atmospheric CO2 might be used as a method to detect changes in fossil fuel emission. We re-read the abstract and found it to be clear. Thus, no changes were made to the text.

2. *Line 50, Newmant et al., (2016, ACP) also studied the fossil fuel section based on the observation approach.*

   **Authors:** Thanks for pointing this out**.** This reference was added to the text

3. *Line 63, it can't be true that the model dynamic is fully resolved. The subgrid processes should be parameterized.*

   **Authors:** This was indeed not carefully worded. Of course, no model can fully resolve all scales of motion. In response, we change the sentence to "A key advantage of this set of approaches is that the spatiotemporal dynamics is fully resolved to the limit provided by the resolution of the transport model."

4. *Line 87-93, the authors is explaining the possibility to detect the fossil fuel signal, which I was assuming that it referred CO2 concentration. However, the two approaches – bottom up and top-down – are used to quantify CO2 fluxes in general. This confusion appeared a lot throughout the entire paper and needs to be fixed.*

   **Authors:** Here, in fact, we are referring to methods that either detect the signal (concentration) or the fluxes. In response, we changed the sentence to
   "In fact, several studies already explored the possibilities to detect the fossil fuel signal and the emissions driving them."

5. *Line 132, in the model setup, the flux components are called surface flux and, can be*

*considered as surface boundary condition. The prescribed global model output providing the advection on the boundaries of model domain is lateral boundary conditions. The statement regarding these drivers is not clear to me.*

**Authors:** Yes, the description of how we treated the surface and lateral boundary conditions was not entirely clear (see also main comment by the 1st reviewer). In response we have added quite some text to this effect. In addition, we added "surface" to the specific line mentioned by this reviewer to read:

" we employ a regional high-resolution atmospheric transport model for the European domain and prescribe lateral and surface boundary conditions for the various components that constitute atmospheric $CO_2$".

*6. Line 170, the authors used time function to scale the monthly emission in time. I recommend the authors to make a figure that shows the time series of the scaled emission components and the total emission on top of Figure 2a. I also would like to see how the authors reconcile the discontinuity between weeks and between months. (plot the month and weeks?)*

**Authors:** It seems as if we had confused the reviewer regarding our approach. Our starting point are the annual totals (line 171). These annual totals were then scaled with the product of three time functions, i.e., f(t) = f_hour(t) * f_week(t) * f_season(t), where f_hour is the diurnal time function, f_week is the weekly one, and f_season the seasonal one. As these are time-continuous functions, there is no discontinuity between weeks and months. In order to make our approach clearer, we added this equation to the methods. Namely, the text now reads:

"The scaled emission flux (E(t)), was then given by

$E(t) = E_{ann} * f_{hour}(t) * f_{week}(t) * f_{season}(t)$,

where t is the time, $E_{ann}$ is the annual integrated fossil fuel emission, $f_{hour}(t)$ is the diurnal time function, $f_{week}(t)$ is the weekly one, and $f_{season}(t)$ is the seasonal one."

We are also not entirely sure about how to respond to the request by the reviewer to add the totals, since this is what is already shown in Figure 2b, albeit for a few countries only. What we propose to do is to add two plots showing the average emission density for the domain, as well as that for a few

more countries (see below). Since we computed the impact of the sectorial emissions with time-constant emissions, it makes no sense to show these scaled emissions.

[Figure]

*Proposed modified Figure 2: We have added the total emissions for the domain (panel c), and for the three additional countries/regions (panel d).*

*7. Line 207, a table to describe all of the fossil fuel tracers will be very helpful for readers to follow.*

**Authors:** We propose to add the following table to the text.

| Name | Character | Time function |
|------|-----------|---------------|
| *CO2_C_tot* | *Tracer of total emissions* | constant |
| *CO2_C_he* | *Tracer of emissions from heating* | constant |
| *CO2_C_in* | *Tracer of emissions from industry* | constant |

| | | |
|---|---|---|
| *CO2_C_pp* | *Tracer of emissions from power plants* | constant |
| *CO2_C_ro* | *Tracer of emissions from road transport* | constant |
| *CO2_C_re* | *Tracer of emissions from other sources* | constant |
| *CO2_p_tot* | *Tracer from total emissions* | Time varying |
| *CO2_p_CH* | *Tracer of emissions from Switzerland* | Time varying |
| *CO2_p_GE* | *Tracer of emissions from Germany* | Time varying |
| *CO2_p_FR* | *Tracer of emissions from France* | Time varying |
| *CO2_p_IT* | *Tracer of emissions from Italy* | Time varying |
| *CO2_p_AU* | *Tracer of emissions from Austria* | Time varying |
| *CO2_p_NL* | *Tracer of emissions from the Netherlands and Belgium* | Time varying |
| *CO2_p_UK* | *Tracer of emissions from the United Kingdom* | Time varying |
| *CO2_p_SW* | *Tracer of emissions from southwestern countries (Spain and Portugal)* | Time varying |
| *CO2_p_EA* | *Tracer of emissions from eastern European countries* | Time varying |
| *CO2_p_other* | *Tracer of emissions from other regions (e.g. maritime emissions by shipping)* | |

*8. Line 207 and 226 contradict each other. Please double check how many fossil fuel tracers used, and correct it if it's not.*

**Authors**: 17 is the correct number. We thus changed "15" into "17" in Line 226

*9. Line 225, see item 5 above.*

**Authors**: we changed "the lateral and boundary conditions" to "the lateral and surface boundary conditions".

*10. Line 235, does R>0.7 mean significant correlation?*

**Authors**: In these comparisons, any correlation with an R>0.7 is highly significant. We adjusted the text by adding this information.

*11. Line 248, "underestimate" or overestimate? All of the bias values in Table 1 are positive but the one at Mace Head.*

**Authors**: Thanks for spotting this mistake. It should be "*overestimate*". But please note that since we changed the basis of our evaluation, all numbers in Table 1 have changed.

*12. Line 248 – 259, I can't find those value in the Table. It's confusing.*

**Authors**: We changed the sign in table 1.

*13. Line 261, in Table 1 the STD for the model are ~4 ppm and ~9ppm at Mace Head and Hungary, respectively. Please correct them and carefully check the values are stated in the manuscript.*

**Authors**: They are correct, because we are describing the model results in Line 261. But please note that since we changed the basis of our evaluation, all numbers in Table 1 have changed.

*5.      Line 300, see item 10.*

**Authors**: Compared to biases from other models, e.g., those from Bozhinova et. al., (2014), these comparisons are good to excellent. We thus consider our statement as warranted.

*15. Line 314, what is the criteria for "the good to excellent evaluation"? Could the authors quantify them?*

**Authors**: (see also item 14) Of course, such statements are always relative to an expectation formed by how well previous modeling studies were able to fit the observations. Using the results of Bozhinova et. al., (2014), as a benchmark, our

results are indeed very encouraging, leading us to conclude that our model is doing a good to excellent job. Nevertheless, in response to this comment, we deleted "good to excellent"

*16. Line 317 – 319, "It is particularly ....". " The presence of an overall ..." These two sentence contradict each other.*

**Authors**: We deleted the sentence "It is particularly encouraging to note the good agreement not only for the fossil fuel $CO_2$ component, but also for total atmospheric $CO_2$. "

*17. Line 320, the statistics were made sometimes based on the daytime values sometimes based on the all time series. Overall, the authors did model evaluation (comparison to observations) based on the daytime values but did the annual mean based on the all time series. The models are struggled with simulating the nighttime CO2 in general. The all time series will very likely bias the entire results of annual mean.*

**Authors**: We refer back to our response above. We do not share this reviewer's concern about his/her perceived inconsistency between the evaluation and the results section. But we agree that it is somewhat awkward to use two different averaging periods. We thus now show true daily means for both evaluation and results.

In response, we added a sentence to this effect to the results section. It reads: "We computed this mean using data from all times of the day in order to fully reflect the annual mean."

*18. Line 333, more explanation needed on "suggesting a somewhat limited efficiency of atmospheric transport and mixing to disperse the signal laterally". More explanation needed.*

**Authors**: What was meant was that the emissions tend to get trapped near their sources, i.e., that the transport is not very effective in mixing these signals aloft and in lateral directions. In order to reflect this, we reformulated the sentence to "...suggesting a somewhat limited effectiveness of atmospheric transport and mixing to disperse the signal aloft and in lateral directions".

*19. Line 334, a map of terrain for the region of interest will be very helpful.*

**Authors**: We will add the following topographic map of the model to the supplementary section of the paper.

[Figure]

*20. Line 360, "while the biospheric signal is stronger". Biospheric signal can be positive and negative, which will cause completely opposite effect on the total CO2. Please clarify it.*

**Authors**: We reformulated these sentences to read "At the same time, the biospheric signal changes sign in the south and becomes positive. This compensates for the smaller fossil fuel signal there and results in a relatively uniform spatial pattern of atmospheric $CO_2$ across Europe". We hope this clarifies this.

*21. Line 369, the column averaged values are smaller than the surface ones, mainly because the signals are at surface and they are diluted in the column as averaged out. The statement is incorrect.*

**Authors**: The reviewer is correct that the dilution of the surface signals with the air aloft leads to the lower concentration. Our statement does not contradict this at all. It rather emphasizes why the concentrations aloft are smaller. One element that we had not mentioned before is the role of the lateral boundary

conditions, whose role is more important aloft than at the surface. We thus revised our statement accordingly. It now reads:

"An additional reason is a much stronger influence of the lateral boundary conditions, which result in a dilution of the fossil fuel components."

*22. Line 400, what does lateral gradient mean specifically here?*

**Authors**: It refers to the spatial gradient in horizontal direction. We believe that the use of the expression "lateral" is accurate and we thus decided to keep it.

*23. Line 419-428, Figure 8 and the relevant statement doesn't make sense to me. I don't understand that why the contribution of each component (b-d) would be larger than the total (a).*

**Authors**: This - at first surprising - effect can arise as a result of potentially compensating effects between the different components. This happens when the different components are anti-correlated. To clarify this, we added a short statement to the text. It reads:

"To determine the contribution, we then computed the fractional variance of each component relative to the total variance. Since the different temporal components can compensate for each other, the sum of the fractional variance can actually exceed unity."

*24. Line 487-555, the authors made misleading statement regarding the relationship between meteorological matric (such as PBL, etc.) and fossil fuel emission. Fossil fuel emission is strongly correlated to the anthropogenic activities. The increase and decrease of the fossil fuel emission is not affected by transport and/or mixing; the atmospheric co2 concentration is. At the beginning, I thought it ("emission") is a typo, which is an easy fix. However, I realized that this relationship is not clear at all to the authors when I saw Figure 9b.*

**Authors**: It appears as if we confused the reviewer. We are, of course, not implying a causation between atmospheric transport and emissions. Rather we are implying that their correlation causes a net signal in atmospheric $CO_2$ in the presence of a net zero flux. This phenomenon, coined rectification effect by Denning et al., (1995) has been thoroughly discussed in the literature. Most of the literature was concerned about the terrestrial rectification, arising on diurnal and seasonal timescales from the correlation between atmospheric

(vertical) mixing and the net exchange fluxes of the terrestrial biosphere, but more recently, this term has also been used for fossil fuels (e.g. Zhang et al., 2016). We re-read what we have written, and could not identify where we went wrong. Thus, no changes were made to the text.

*25. Line 670, the authors selected three location to do further analysis for Figure 15. Why these three? Please explain.*

**Authors**: The exact locations of these three locations were chosen somewhat arbitrarily, but were identified on the basis of us seeking examples of the three different cases of interest. (i) The changes in the average concentration are larger than those of the standard deviation, (ii) the changes in the average are of similar magnitude as those of the standard deviation, and (iii) the changes in the average are smaller. We clarified this in the text.

*26. Line 700, yes, the authors were being optimistic as they admitted. All of the analyses were based on the model results. The cloud contamination was considered in the discussion. However, the more important errors are very likely caused by model transport errors and error estimate of biogenic flux. It will be great if the author can expend the discussion on the impact of those error on their findings.*

**Authors**: We absolutely agree. In response we added the following two sentences to the text "We assumed here also "perfect transport", i.e., no errors in how the emission reductions manifest themselves in a change in the concentration field. In fact, errors in this transport are, perhaps, next to the lack of observations that largest impediment to detect changes in fossil fuel emissions."

*27. Line 724, where does "110%" come from? Is it a new? The authors shouldn't include any new results in the conclusions.*

**Authors**: Thanks for pointing this out. While this result is actually not new, we nevertheless deleted it.

*28. Figure 1, please put the labels on the color bars.*

**Authors**: *added as requested.*

*29. Figure 2, I would like to see the time series of the scaled emission components and the total emission. (plot total scaling factors)*

**Authors**: See our reply to point 6 above.

*30. Figure 4, 1) enlarge the font size. 2) what year is it? 3) night time model results included? If so, 10 m above ground is too late. Models are struggled at nighttime, especially for such low level. 4) Keep the color bars in the scale and same range for a and c, b and d.*

1) *enlarge the font size.*
   **Authors**: changed
2) *what year is it?*
   **Authors**: The whole paper is based on simulations from one year, as stated in the method part, i.e., from end of March 2008 till March 2009
3) *night time model results included?*
   **Authors**: Yes.
4) *Keep the color bars in the scale and same range for a and c, b and d.*
   **Authors**: We use the same scale for a and c, b and d, and we keep the same range for all the 4 figures.

*31. Figure 5, the figure labels (a, b, c, d) are missing.*

   **Authors**: Added to the figure.

*32. Figure 6, use the same color scale as Figure 4 does.*

   **Authors**: As Figure 6 shows a different quantity than Figure 4 (column averaged $XCO_2$ instead of the surface mixing ratio), and a quantity whose spatial gradients are much smaller, we decided to the keep range. This advantage clearly outweighs the benefit being able to directly compare Figures 4 and 6.

*33. Figure 8 doesn't make sense to me as I pointed out before. The contribution of each component can't larger than the total.*

   **Authors**: Please see our explanation above.

*34. Figure 9a, which minus which? Figure 9b doesn't make sense to me. The fossil fuel emission is related to anthropogenic activities; it won't be affected by transport and/or mixing. If it is concentration instead of emission. The correlation of the concentration and PBL height should be negative. It's confusing.*

**Authors**: Please see our explanations above. We are indeed concerned here about the temporal correlation between fossil emissions and vertical mixing/transport. But please recall that a correlation does not imply a causation. We clarified in the caption the sign of the signal, i.e., that the plot is showing the difference between the time varying and the time constant emission case, i.e., "(time varying minus time constant)". Figure 9b is correlation between the emission and PBL height. This figure just helps us to explain the map in Figure 9 a.

*35. Figure 11, what countries are they exactly included in SW?*
   **Authors**: Portugal and Spain. We clarified this in the caption.

*36. Table 1, it's a confusing statement that "the 3 hourly means between 12 – 18 PM"*

   **Authors**: It is indeed confusing. We thus changed "the 3 hourly means between 12 – 18 PM" changed into "the mean between 12 and 18 PM based on the 3 hourly output of the model."

*37. Figure 13, enlarge the font size. Does it in percentage? Can the authors overlay the wind field on top?*

   **Authors**: We changed the font size. Yes, the numbers refer to percentage, as we had stated in the caption "relative contribution". As the figures contain already too much information, we decided against the overlaying of the wind field.

*38. Figure 14, it's not clear to me to look at the standard deviation changes between two cases.*

   **Authors**: The idea here is analyze whether changes in the variability of atmospheric $CO_2$ can be used to detect changes in fossil fuel emissions in addition to changes in the mean concentration. However, Figure 14 suggests that this method might be not so promising when using the column averaged $CO_2$.

*39. Figure 15, I can only find three green dots in Figure 14b, but Figure 15b shows four lines. Please make correction accordingly.*

   **Authors**: Thanks for spotting this. The two dots (corresponding to blue and cyan line) were overlaying each other due to their short distance. We changed this in the figure by using opacity.

**References**:

D. Bozhinova, M. K. Van Der Molen, I. R. van der Velde, M. C. Krol, S. van der Laan, H. A. J. Meijer, and W. Peters. Simulating the integrated summertime $\Delta 14CO2$ signature from anthro- pogenic emissions over Western Europe. Atmos. Chem. Phys., 14(14):7273–7290, 2014

Houweling, S., Aben, I., Breon, F.-M., Chevallier, F., Deutscher, N., Engelen, R., Gerbig, C., Griffith, D., Hungershoefer, K., Macatangay, R., Marshall, J., Notholt, J., Peters, W., and Serrar, S.: The importance of transport model uncertainties for the estimation of CO2 sources and sinks using satellite measurements, Atmos. Chem. Phys., 10, 9981-9992, doi:10.5194/acp-10-9981-2010, 2010.

F. R. Vogel, B. Thiruchittampalam, J. Theloke, R. Kretschmer, C. Gerbig, S. Hammer, and I. Levin. Can we evaluate a fine-grained emission model using high-resolution atmospheric transport modelling and regional fossil fuel CO2 observations? Tellus B, 65:1–16, 2013. doi: 10.3402/tellusb.v65i0.18681.

---

## Referee Report (RR1)

**Review of Liu et al.: Spatiotemporal patterns of the fossil-fuel CO2 signal in central Europe: Results from a high-resolution atmospheric transport model (R2)**

General comments: the authors have addressed most of the comments that I had for the first round of the review. However, there are still some points that are not clear to me. I would recommend the manuscript to be published after the authors address the comments below.

Major comments:

- Time functions:

  I have pointed out this issue in the first round of the review. The authors have added Equation 2 in the revised manuscript correspondingly. First of all, in Eq.1, what is the range and total number of "t"?

  The mass conservation of carbon is critical in the downscaling approach. At the same time, simple downscaling approach using time functions likely leads the "stair-stepping" behavior between months (See Figure 1 in Fisher et al., 2016). Can the authors zoom in Figure 2 just to show the transit between two months to see if the "stair-stepping" behavior exists in the approach they applied? If it doesn't, I would like to know more details about how the authors reconcile this behavior. If it does, I would like to see more discussion on how this issue affects/biases the results.

- In the revision, the authors have done all of the analysis based on the full time series of the period of interest to have the consistency for the entire manuscript. However, it is well known and also showed in the manuscript that the nighttime CO2 signals are much larger than the daytime ones. In the section of discussion, the authors have discussed the availability of the detection of the satellite measurements to the reduction of 30% of the fossil fuel. Apparently, the authors understand that the existing CO2 satellite (e.g., GOSAT and OCO-2) sample CO2 around 1pm local time. In the case, further discussion based on the results of the full time series is not appropriate any more. Although the result won't change the main conclusion (the gradient and variability of XCO2 will be even less according to Figure 14), cautious clarification is needed here.

  Minor comments:

1. Line 65, removed "fully". I still don't think "fully" can be used in this context.

2. Line 270 and below, apparently, the correlation between observation and model results becomes smaller after the authors use the full time series instead of daytime only. Do all of the values the authors listed pass the significance test at $P>0.005$? I am suspicious about 0.57 and 0.63. Can the author clarify it?

3. Line 515, can the authors briefly explain what the "other factors" are? And how?

4. Figure 8, a different method is needed here to present the contribution of each component to the total. It is confusing to me the total contribution of each could be larger than the total (a).

References:

Fisher, J. B., Sikka, M., Huntzinger, D. N., Schwalm, C., and Liu, J.: Technical note: 3-hourly temporal downscaling of monthly global terrestrial biosphere model net ecosystem exchange, Biogeosciences, 13, 4271-4277, https://doi.org/10.5194/bg-13-4271-2016, 2016.

---

## Author Response (AR2)

**Spatiotemporal patterns of the fossil-fuel $CO_2$ signal in central Europe: Results from a high-resolution atmospheric transport model**

Yu Liu, Nicolas Gruber, and Dominik Brunner

We would like to thank both reviewers for their efforts to re-review our manuscript. While reviewer #1 was happy with the revision and had no further comments, reviewer #2 had a few more requests/comments that we address here point by point.
* * *
**Referee #2**

**General comments:**

*Referee#2: Time functions:*
*I have pointed out this issue in the first round of the review. The authors have added Equation 2 in the revised manuscript correspondingly. First of all, in Eq.1, what is the range and total number of "t"?.*

**Authors:** In Eq.1, the meaning of $t$ is hour of the year. To make it clear, we added the following:

"where t is time (hour of the year) and $f_{diurnal}$, $f_{week}$ and $f_{season}$ are the diurnal, weekly, and seasonal scaling factors, respectively.

*Referee#2: The mass conservation of carbon is critical in the downscaling approach. At the same time, simple downscaling approach using time functions likely leads the "stair-stepping" behavior between months (See Figure 1 in Fisher et al., 2016). Can the authors zoom in Figure 2 just to show the transit between two months to see if the "stair-stepping" behavior exists in the approach they applied? If it doesn't, I would like to know more details about how the authors reconcile this behavior. If it does, I would like to see more discussion on how this issue affects/biases the results.*

**Authors:** We agree that mass conservation is important, but this is simply ensured by the fact that the mean of the product of $f_{diurnal} \cdot f_{week} \cdot f_{season}$ is set equal to 1. We also attempted to avoid as much stair-stepping as possible, primarily by linearly interpolating between the months. In order to demonstrate this, we zoom in the Figure below (Figure 1) into two months, i.e., April and May. On top of the very strong diurnal cycle, the weekly rhythm of emissions is clearly visible with the substantially lower emissions during the weekend. Also seen is the smooth seasonal transition from April to May. Thus, we cannot find any evidence for stair-steps in our time function.

[Figure]

*Figure 1: Temporal evolution of the emissions for a two-month long section in April and May.*

While we do not think that we have to demonstrate this with a figure in our paper, we agree that some further text elaborating on our scheme is helpful. In response, we will add the following text to the paper:

"The time function factor $f_{diurnal}$ depends on the hour of the day (local time, $t_{hour} = t$ modulo 24h) and is different for weekdays and weekends to reflect the different level of activities on weekdays and weekends. The factor $f_{week}$ depends on the day of the week, with one value for weekdays (Monday-Friday) and a lower one for Saturdays and Sundays. The factor $f_{season}$ depends on the month, but in order to avoid discontinuities between subsequent months, it is linearly interpolated to a given day between the centers (day 15) of two adjacent months."

*Referee#2: In the revision, the authors have done all of the analysis based on the full time series of the period of interest to have the consistency for the entire manuscript. However, it is well known and also showed in the manuscript that the nighttime CO2 signals are much larger than the daytime ones. In the section of discussion, the authors have discussed the availability of the detection of the satellite measurements to the reduction of 30% of the fossil fuel. Apparently, the authors understand that the existing CO2 satellite (e.g., GOSAT and OCO-2) sample CO2 around 1pm local time. In the case, further discussion based on the results of the full time series is not appropriate any more. Although the result won't change the main conclusion (the gradient and variability of XCO2 will be even less according to Figure 14), cautious clarification is needed here.*

**Authors:** We agree with the reviewer that it does not make sense to base the discussion of the detection of plumes by satellites on the basis of the diurnally averaged data. This is why we used the results at 1 pm, not the full time series for this section. This was not adequately mentioned in the revised version of the manuscript. We thus will ensure that this is entirely clear in the new version by adding the following in the text: "As the satellites have a typical overpass time of 1:00 PM local time, we conducted all subsequent analyses using the model data only from this time slot."

**Specific comments:**

1.      *Line 65, removed "fully". I still don't think "fully" can be used in this context.*
**Authors:** Done

*2.       Line 270 and below, apparently, the correlation between observation and model results becomes smaller after the authors use the full time series instead of daytime only. Do all of the values the authors listed pass the significance test at P>0.005? I am suspicious about 0.57 and 0.63. Can the author clarify it?*

**Authors:**   We checked all the correlations and they pass the significance test at $p<0.05$, i.e., at the level indicated in the text (We suspect that the reviewer also meant 0.05 instead of 0.005). However, we noted a mistake of ours in that we wrote "at the $p>0.05$ level" instead of "at the $p<0.05$ level". This is now corrected.

*3.       Line 515, can the authors briefly explain what the "other factors" are? And how?*

**Authors:** As this expression does not occur on line 515, but on line 491, we suspect that the reviewer was referring to this place in the manuscript. With other factors, we meant the diurnal variations in atmospheric transport and mixing. But this is already mentioned further down in this paragraph. We thus deleted this statement.

*4.Figure 8, a different method is needed here to present the contribution of each component to the total. It is confusing to me the total contribution of each could be larger than the total (a).*

**Authors:** This is certainly not immediately intuitive, but any other relative measure would distort the different contributions. Thus, we are clearly of the opinion that this is the best and most correct representation of what we would like to convey. Nevertheless, it is probably a good idea to guide the reader a bit more. Thus we added the following statement to the caption of figure 8.

[revised manuscript text omitted]